# Mitochondrial network configuration influences sarcomere and myosin filament structure in striated muscles

Prasanna Katti [1], Alexander S. Hall[2], Hailey A. Parry[1], Peter T. Ajayi [1], Yuho Kim [1], T. Bradley Willingham[1], Christopher K. E. Bleck [1], Han Wen [1] & Brian Glancy [1,3] ✉

Sustained muscle contraction occurs through interactions between actin and myosin filaments within sarcomeres and requires a constant supply of adenosine triphosphate (ATP) from nearby mitochondria. However, it remains unclear how different physical configurations between sarcomeres and mitochondria alter the energetic support for contractile function. Here, we show that sarcomere cross-sectional area (CSA) varies along its length in a cell type-dependent manner where the reduction in Z-disk CSA relative to the sarcomere center is closely coordinated with mitochondrial network configuration in flies, mice, and humans. Further, we find myosin filaments near the sarcomere periphery are curved relative to interior filaments with greater curvature for filaments near mitochondria compared to sarcoplasmic reticulum. Finally, we demonstrate variable myosin filament lattice spacing between filament ends and filament centers in a cell type-dependent manner. These data suggest both sarcomere structure and myofilament interactions are influenced by the location and orientation of mitochondria within muscle cells.

Force generation within striated muscle cells occurs through the formation and release of bonds between actin and myosin filaments within the sarcomere with networks of highly connected sarcomeres[1–4] occupying up to 85% of cellular volume[4–6]. Sustained muscle contractions also necessitate continued provision of calcium from the sarcoplasmic reticulum (SR) and adenosine triphosphate (ATP) from mitochondria and/or glycolytic enzymes, thereby requiring significant cellular volume in proximity to the sarcomeres to be invested in these systems as well[7].

The chronic force production demand (i.e., magnitude and frequency) on a given muscle cell type is well known to influence the amount of cellular volume allocated to, and thus the functional capacity of, each component[7–16]. However, it is less well understood how variations in contractile demand affect the spatial organization among the tightly packed sarcomeres and the organelles that must fit in between them or how different cellular architectures alter the

functional support for muscle contraction. In particular, while mitochondria are known to form cell-type-dependent networks aligned perpendicular and/or in parallel to the contractile axis of the muscle cell[11,17–20], the functional implications of placing a mitochondrion in parallel between sarcomeres versus wrapping a mitochondrion perpendicularly between sarcomeres near the Z-disks remains unclear.

Muscle cell architecture is well known to be driven by muscle fiber type[21,22]. In mammals (e.g., human or mouse), muscle fiber type is commonly defined based on contractile type (e.g., fast- or slow-twitch) and/or metabolic type (e.g., oxidative or glycolytic)[23,24], though hybrid muscle cells that co-express multiple isoforms of the myosin heavy chains which determine contractile type can exist in significant populations[25,26]. Oxidative muscles containing large (~10–15% cell volume), grid-like mitochondrial networks arranged both perpendicular and parallel to the contractile apparatus can be either fast- or slow-twitch in nature, whereas glycolytic muscles with smaller (~2–3%

[1]National Heart, Lung and Blood Institute, National Institutes of Health, Bethesda, MD, USA. [2]Thermo Fisher Scientific, Houston, TX, USA. [3]National Institute of Arthritis and Musculoskeletal and Skin Diseases, National Institutes of Health Bethesda, Bethesda, MD, USA. ✉e-mail: Brian.glancy@nih.gov

cell volume) mitochondrial networks primarily aligned perpendicularly are typically only found in fast-twitch fibers[7].

In insects (e.g., the fruit fly *Drosophila melanogaster*), muscle type has generally been defined based on contractile (e.g., fibrillar or tubular) and/or electromechanical (e.g., synchronous or asynchronous) characteristics[27–31]. The commonly studied *Drosophila* indirect flight muscle has a fibrillar contractile type due to its individual myofibrils which run the entire length of the cell and is considered asynchronous since propagation of a single electrical pulse can result in tens of muscle contractions[29,32,33]. While both of these characteristics are unlike any known mammalian muscle, the very large mitochondrial networks (>30% cell volume) arranged in parallel to the contractile apparatus within indirect flight muscles appear similar to those in mammalian cardiac cells. Conversely, the majority of *Drosophila* muscles are synchronous (one electrical pulse results in one contraction) and have a tubular contractile type[34] consisting of branched myofibrillar networks similar to mammalian muscles[4].

Though there is relatively little information available regarding the variability in metabolism across tubular muscles[35], recent work has shown *Drosophila* tubular muscles can vary in mitochondrial content from ~2–3% to greater than 20% of cell volume[4] with mitochondrial networks arranged in either parallel or grid-like configurations[36]. Additionally, transcription factors involved in the determination of fly muscle cell fate such as *salm*[29], *cut*[37,38], and *H15*[36], are now also known to regulate mitochondrial network structure in both tubular and fibrillar muscles[36,39]. However, it remains unclear in both mammalian and insect muscles how altering mitochondrial network configuration may impact the structure/function relationships of adjacent subcellular structures.

Increasing mitochondrial volume in striated muscle largely comes at the expense of the contractile apparatus[40]. Thus, we aimed to determine how displacement of contractile volume is coordinated with the varying mitochondrial network configurations that occur across different muscle cell types[11,17–20,36,39]. We hypothesized that mitochondria arranged in long, parallel rows between the sarcomeres effectively displace contractile volume uniformly along the length of the sarcomere, whereas mitochondria wrapped perpendicularly near the sarcomere ends only require local displacement of sarcomere structure in order to maximize efficient utilization of limited cellular volume. Therefore, we predicted that sarcomeres with adjacent, perpendicularly aligned mitochondria would have a smaller cross-sectional area (CSA) at the sarcomere ends (Z-disks) relative to the CSA in the middle of the sarcomere (M-line or A-band). To test this prediction, we investigated the detailed physical interactions among sarcomeres and their adjacent organelles in eleven muscle types with varying contractile and mitochondrial network configurations across three species (human, mouse, fly) using high resolution, 3D volume electron microscopy.

Here, we demonstrate that sarcomere cross-sectional area (CSA) in mice, flies, and humans is smaller at the Z-disk ends than in the sarcomere centers where myosin resides. Further, we find that the magnitude of intrasarcomere CSA heterogeneity is cell type-dependent and closely coordinated with the location of mitochondria between the sarcomeres rather than the total volume or size of mitochondria within the muscle cell. By performing a massively parallel myosin filament analysis in mouse and fly muscles, we show that intrasarcomere CSA heterogeneity is achieved, at least in part, by curvature of the contractile filaments near the periphery, but not the core, of each sarcomere. Across muscle cell types, the magnitude of myosin curvature is highest in cells with large proportions of mitochondria running perpendicularly near the Z-disk, while within cells, myosin curvature is greater for filaments near mitochondria and lipid droplets compared to the sarcoplasmic reticulum. Moreover, intrasarcomere heterogeneity in myosin shape results in variable filament-to-filament lattice spacing along the length of the sarcomere where

myosin filaments are closer together near the filament ends than in the middle in mice. Additionally, we demonstrate that both myosin filament linearity and intrasarcomere lattice spacing are regulated in *Drosophila* by muscle and mitochondrial network specification factors *salm*[29,36,39] and *H15*[4,36]. Finally, we show that acute swelling of mitochondria in mouse skeletal muscle results in increased curvature of adjacent myosin filaments. Together, these data indicate that sarcomere and myosin filament structure are influenced by where mitochondria are placed within a muscle cell.

## Results

### Sarcomere cross-sectional area varies along its length

To better understand how muscle cell architecture relates to muscle cell function, we performed a 3D survey of sarcomere and adjacent organelle structures across eleven muscle types whose contractile functions have generally been well defined[1,27,41–44]. We performed focused ion beam scanning electron microscopy (FIB-SEM) with 5–15 nm pixel sizes on cardiac and skeletal muscle tissues fixed in vivo in mice and on skeletal muscle tissues from *Drosophila melanogaster* (18 FIB-SEM datasets available at https://doi.org/10.5281/zenodo.5796264).

Close inspection of a mouse fast-twitch glycolytic muscle raw FIB-SEM volume (Fig. 1a, Supplementary Movie 1) suggested that sarcomere cross-sectional area (CSA) was variable at different regions along its length with the CSA becoming smaller near the Z-disk ends compared to the center of the sarcomere within the A-band (myosin containing region). Indeed, 3D rendering of the fast-twitch glycolytic muscle sarcomeres revealed large gaps between the parallel sarcomeres near the Z-disks (Fig. 1b, Supplementary Movie 2) and these gaps corresponded to where perpendicularly aligned mitochondria (Fig. 1c–e, g, h) and SR doublets (Fig. 1d, h) were located adjacent to the sarcomere. Conversely, *Drosophila* indirect flight muscles (IFM) appear to maintain a constant CSA across the entire sarcomere with no large gaps apparent between parallel sarcomeres (Fig. 1i, Supplementary Movie 3) despite a large volume of adjacent mitochondria (Fig. 1j). Thus, we hypothesized that intrasarcomere CSA heterogeneity was cell type-dependent.

To test this hypothesis, we compared the maximal CSA at the Z-disk and corresponding A-band for each half sarcomere sheet (Supplementary Movie 4) within muscle volumes from six mouse muscle types and four *Drosophila* muscle types, as well as in FIB-SEM datasets ($30 \times 30 \times 30$ nm pixels) of human vastus lateralis muscle biopsies (Supplementary Movie 5) previously published as part of the Baltimore Longitudinal Study of Aging[19]. The magnitude of intrasarcomere CSA heterogeneity was highly variable across cell types (Fig. 1k) with the lowest values observed in *Drosophila* IFM and jump muscles (4.7 ± 2.4% CSA difference between A-band and Z-disk, $n = 3$ cells, 56 half sarcomere sheets and 9.5 ± 0.8% difference, $n = 3$ cells, 84 half sarcomere sheets for IFM and jump muscles, respectively) and mouse cardiac muscles (10.9 ± 1.8% difference, $n = 5$ cells, 122 half sarcomere sheets). The greatest degree of intrasarcomere CSA heterogeneity was found in the *Drosophila* leg muscles (40.3 ± 6.5% difference, $n = 5$ cells, 21 half sarcomere sheets), mouse oxidative muscles (34.4 ± 4.8% difference, $n = 4$ cells, 86 half sarcomere sheets, and 31.1 ± 1.7% difference, n = 4 cells, 84 half sarcomere sheets for fast- and slow-twitch muscles, respectively) and human muscles (29.3 ± 1.6% difference, $n = 3$ cells, 80 half sarcomere sheets). The wide range of intrasarcomere CSA heterogeneity values across cell types in both fly and mouse muscles, as well as the high values observed in humans, suggest that cell type specific intrasarcomere CSA heterogeneity is conserved from invertebrate to human muscles.

To test whether intrasarcomere CSA heterogeneity is simply an artifact of the fixation and sample processing required to collect the FIB-SEM datasets described above, we performed stimulated emission depletion (STED) super-resolution microscopy on live mouse flexor

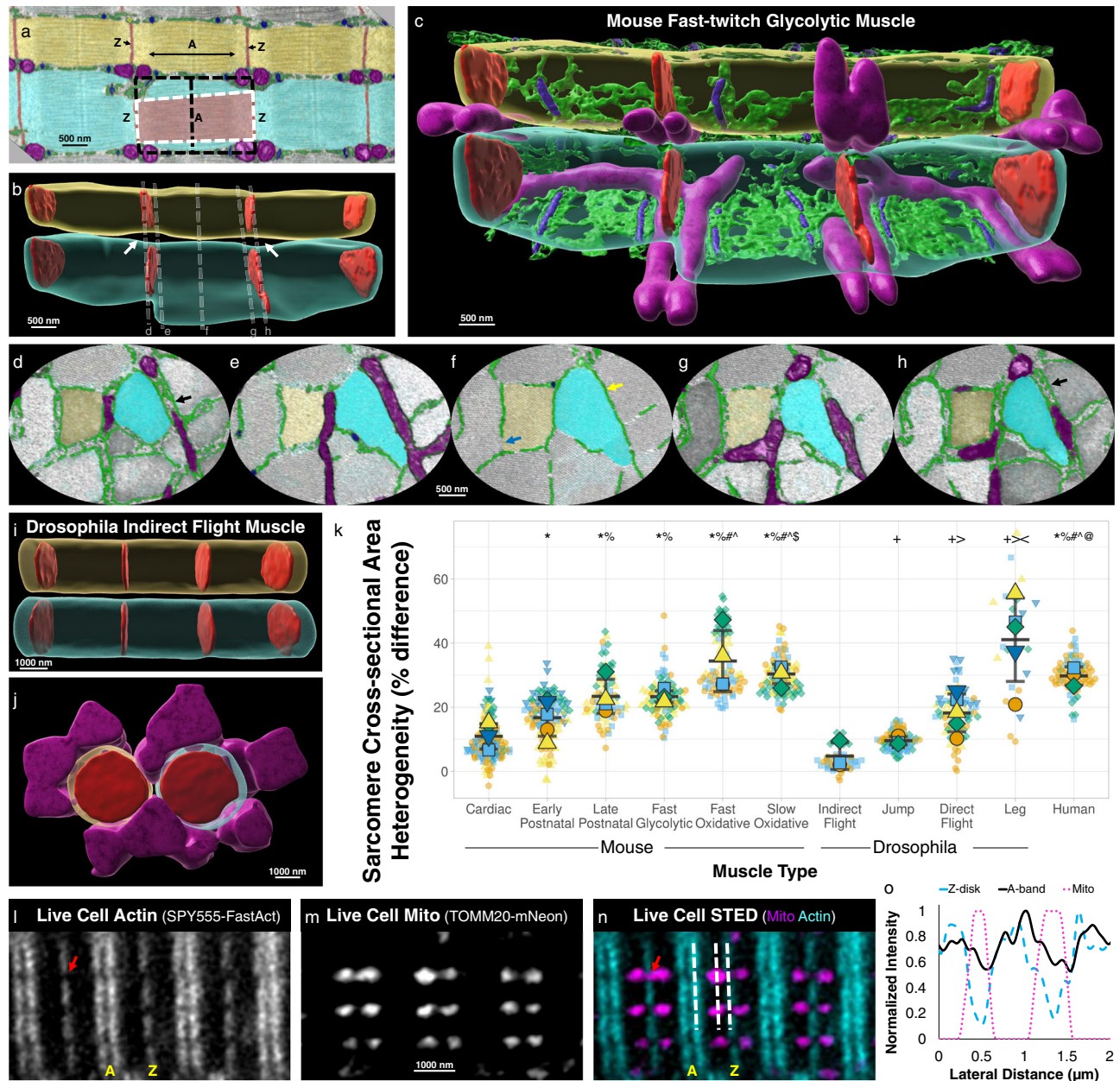

digitorum brevis (FDB) muscles labeled with fluorophores for mito-chondria (genetically encoded TOMM20-mNeon[45]) and F-actin (SPY555-FastAct) (Fig. 1l–n). SPY555-FastAct dye is based on the F-actin linker jasplakinolide[46], and appears to label the actin filament ends at the Z-disk and the central region of the A-band (Fig. 1l, n) similar to genetically encoded actin probes in muscle cells[47]. At the resolution afforded by STED microscopy (>50 nm), the space between laterally adjacent sarcomeres can be clearly resolved at the Z-disk as evident by the large drop in fluorescence (to below 50% of local maximum) laterally between Z-disks (Fig. 1l, n, o). However, the space between the corresponding A-bands cannot be clearly resolved due to an insufficient drop in fluorescence (stays above 50% of local maximum) in these regions. Thus, in live muscle cells, adjacent sarcomeres are closer together at the A-bands than they are at the Z-disks. Consistently, the space between Z-disks where mitochondria and SR doublets are located can be up to 300 nm in diameter in our FIB-SEM images (black arrows in Fig. 1d, h) whereas the space between A-bands consists of longitudinal SR with diameters of ~50 nm or less (yellow arrow in Fig. 1f) or some regions with no organelles at all (blue arrow in Fig. 1f). These data suggest that intrasarcomere CSA heterogeneity and

its relationship with the shape and location of the organelles in between sarcomeres are not driven by FIB-SEM sample preparation artifacts.

## Sarcomere shape is coordinated with mitochondrial location

The six muscle types with greater than 20% intrasarcomere CSA heterogeneity in Fig. 1 (mouse late postnatal and all three mature muscles, *Drosophila* leg, and human vastus lateralis) are all known to have large proportions of their mitochondrial networks which wrap perpendicularly around the sarcomere adjacent to the Z-disks, whereas the remaining five muscle types with less than 20% heterogeneity all have mitochondrial networks primarily oriented in parallel to the contractile apparatus[11,18,19,36,39,48,49]. Thus, we hypothesized that sarcomere structure is linked to mitochondrial network configuration.

To test this hypothesis, we first used machine learning[50] to segment out mitochondrial network structures within each FIB-SEM dataset[11] (Supplementary Movie 6). To assess mitochondrial network configuration, we further separated the mitochondrial network structures into regions adjacent to the Z-disk (light blue in Supplementary Movies 7, 8) and regions not adjacent to the Z-disk (red in

**Fig. 1 | Intrasarcomere cross-sectional area heterogeneity is dependent on muscle cell type. a** Longitudinal raw focused ion beam scanning electron microscopy (FIB-SEM) image of a mouse fast-twitch glycolytic gastrocnemius muscle showing two parallel myofibrillar segments comprised of three sarcomeres each (cyan and yellow). Diameter of the A-band (A) is larger than the diameter of the Z-disk (Z, red). Mitochondria (magenta), sarcoplasmic reticulum (SR, green), and t-tubules (T, blue) are also shown. Representative of four fast glycolytic FIB-SEM datasets. **b** 3D rendering of the sarcomere boundaries from a (translucent cyan and yellow) and their Z-disk structures (red). White arrows highlight gaps between in parallel sarcomeres. **c** Same 3D rendering as b but also showing sarcomere adjacent mitochondria (magenta), and partial sarcoplasmic reticulum (green) and t-tubule (blue) structures. **d–h** Raw FIB-SEM cross-sectional views at the Z-disk, I-band, and A-band regions indicated by dotted lines in b highlighting the varying diameters of the mitochondrial and SR/T structures located between in parallel sarcomeres (cyan and yellow). Black arrows highlight SR doublets. Yellow arrow highlights longitudinal SR. Blue arrow highlights regions without SR between sarcomeres. Representative of four fast glycolytic FIB-SEM datasets. **i** 3D rendering of the sarcomere boundaries (translucent cyan and yellow) for six sarcomeres from *Drosophila* indirect flight muscle and their Z-disk structures (red). **j** 90° rotation of the indirect flight muscle sarcomeres from i also showing adjacent mitochondria which do not run perpendicular to the contractile axis. **k** Percent difference in sarcomere cross-sectional area at the A-band relative to the Z-disk per half sarcomere sheet across eleven muscle cell types. Large shapes represent individual cell values, smaller shapes represent values per individual half sarcomere sheet. Black lines represent cell Mean ± SD. Graph made with SuperPlotsofData web app[117]. **l** Stimulated emission depletion (STED) microscopy image of a live mouse FDB fiber stained with SPY555-FastAct dye. Red arrow highlights intersarcomere space between Z-disks. Images representative of 9 cells, 2 mice. **m** Corresponding mitochondrial (genetically encoded TOMM20-mNeon) image from (**l**). **n** Corresponding live cell actin and mitochondria merged image from (**l, m**). White dotted lines correspond to locations of intensity profiles in (**o**). **o** Fluorescent intensity profile along laterally adjacent sarcomeres corresponding to the Z-disk (cyan dashed line), A-band (black solid line), and mitochondria (magenta dotted line). *significantly different from Cardiac. ˣsignificantly different from Early Postnatal (postnatal day 1). #significantly different from Late Postnatal (postnatal day 14). ^significantly different from Fast Glycolytic. $significantly different from Fast Oxidative. @significantly different from Slow Oxidative. +significantly different from Indirect Flight. ˃significantly different from Jump. <significantly different from Direct Flight. Two-sided, one-way ANOVA with a Tukey's HSD post hoc test. N values: Cardiac-5 mice, 5 cells, 122 half sarcomere sheets; Early Postnatal-5 mice, 5 cells, 106 half sarcomere sheets; Late Postnatal-4 mice, 4 cells, 70 half sarcomere sheets; Fast Glycolytic-4 mice, 4 cells, 86 half sarcomere sheets; Fast Oxidative-4 mice, 4 cells, 86 half sarcomere sheets; Slow Oxidative-4 mice, 4 cells, 84 half sarcomere sheets; Indirect Flight-3 flies, 3 cells, 56 half sarcomere sheets; Jump-1 fly, 3 cells, 84 half sarcomere sheets; Direct Flight-1 fly, 5 cells, 80 half sarcomere sheets; Leg-1 fly, 5 cells, 21 half sarcomere sheets; Human-3 subjects, 3 cells, 80 half sarcomere sheets.

Supplementary Movies 7, 8). Using a distance from Z-disk threshold of 200 nm longer than the width of the sarcomeric I-band (actin only region) in each dataset resulted in all perpendicularly oriented mitochondria to be included in the Z-adjacent mitochondrial pool (Fig. 2a–k, Supplementary Movie 7).

We then compared the percentage of the total mitochondrial network that is adjacent to the Z-disk versus the magnitude of intrasarcomere CSA heterogeneity for all eleven muscle types across the three species which revealed a significant linear relationship (Fig. 2l, $R^2 = 0.4964$, $p < 0.001$) indicating that greater intrasarcomere CSA heterogeneity is associated with more of the mitochondrial pool being localized to the Z-disk. Comparing intrasarcomere CSA heterogeneity to total mitochondrial content also revealed a statistically significant linear relationship (Fig. 2m, $R^2 = 0.1076$, $p = 0.024$), albeit negative and much weaker than with Z-adjacent mitochondrial abundance. Indeed, in a two component multiple regression model evaluating the impact of both Z-adjacent mitochondrial abundance and mitochondrial content on intrasarcomere CSA heterogeneity, Z-adjacent mitochondrial abundance contributes significantly (standardized beta coefficient 0.709, $p < 0.001$) whereas mitochondrial content does not (standardized beta coefficient −0.005, $p = 0.968$). Additionally, there was no significant correlation between intrasarcomere CSA heterogeneity and individual mitochondrial volume (Fig. 2n, $R^2 = 0.059$, $p = 0.126$).

These data suggest that, from flies to humans, sarcomere shape appears to be closely coordinated with where mitochondria are located within a muscle cell, and less so with how many or how big the mitochondria are. It should also be noted that sarcomere length was not controlled for or specifically modulated in these studies, and no consistent relationship between sarcomere length and intrasarcomere CSA heterogeneity across all muscles or within individual cell types was observed (Supplementary Fig. 1), perhaps due to the variability in mitochondrial location across samples. However, future studies will be needed to fully elucidate the impact of sarcomere length on the relationships between mitochondria and sarcomere shape.

Since there are currently no methods to directly assess forces at the sarcomere or myofilament scale within intact muscle cells, as a first step toward assessing the functional impact of intrasarcomere CSA heterogeneity, we developed a simple geometrical model to simulate the relative change in force production due to the addition of a new mitochondrion wrapping perpendicularly around the sarcomere near the Z-disk (Fig. 2o). Simulations of two types of sarcomeric adaptations to mitochondrial addition were performed. First, sarcomere CSA was maintained constant along its length, as is commonly assumed[51–55], and, thus, reduction of the Z-disk CSA to make space for the mitochondrion results in a proportional loss in the number of linear myofilaments throughout the sarcomere. As expected, simulations of this uniform sarcomere CSA model showed that isometric force production decreased linearly with the magnitude of reduction of the Z-disk CSA (orange dotted line in Fig. 2o).

The second model tested was based on the intrasarcomere CSA heterogeneity structures described above where the Z-disk, but not the middle, of the sarcomere is compressed to make room for the new mitochondrion. It was assumed in the heterogeneous CSA model that myosin filaments curved slightly in proportion to the reduction in Z-disk CSA but that filament number remained constant. In this model, force decreases as the cosine function of the tilt angle of the filaments, with increasing angles occurring in proportion to the reduction in Z-disk CSA. These simulations (blue line in Fig. 2o) revealed a negligible (1.8%) loss of force in response to up to 40% reduction of the Z-disk CSA, similar to the highest mean CSA difference measured in Fig. 1k. However, more significant reductions in force can occur in this model with larger reductions in Z-disk CSA, with a 10.3% loss of force at an 80% reduction in Z-disk CSA.

Overall, the data from these geometrical simulations accounting simply for myosin filament number and angle suggest that by limiting the loss of sarcomere CSA to near the Z-disk, there is little contractile cost to adding a mitochondrion wrapped perpendicularly around a sarcomere. However, it is important to note that this simple geometrical model does not account for any potential changes in myofilament molecular dynamics that may occur within the sarcomere as a result of intrasarcomere CSA heterogeneity. More detailed models accounting for the true 3D myofilament structures and their dynamic molecular interactions within the sarcomere at various sarcomere lengths will be needed to more fully understand the functional impact of intrasarcomere CSA heterogeneity.

**Mitochondrial proximity influences myosin filament curvature**

To better understand the influence of intrasarcomere CSA heterogeneity on the myofilaments within the sarcomere, we performed massively parallel segmentations of all the myosin filaments within our mouse muscle FIB-SEM datasets (Fig. 3a, Supplementary Movies 9, 10) which allowed for assessment of hundreds of thousands to millions of individual filaments per cell. We assessed the accuracy of our myosin filament segmentations by overlaying the skeletons of each segmented

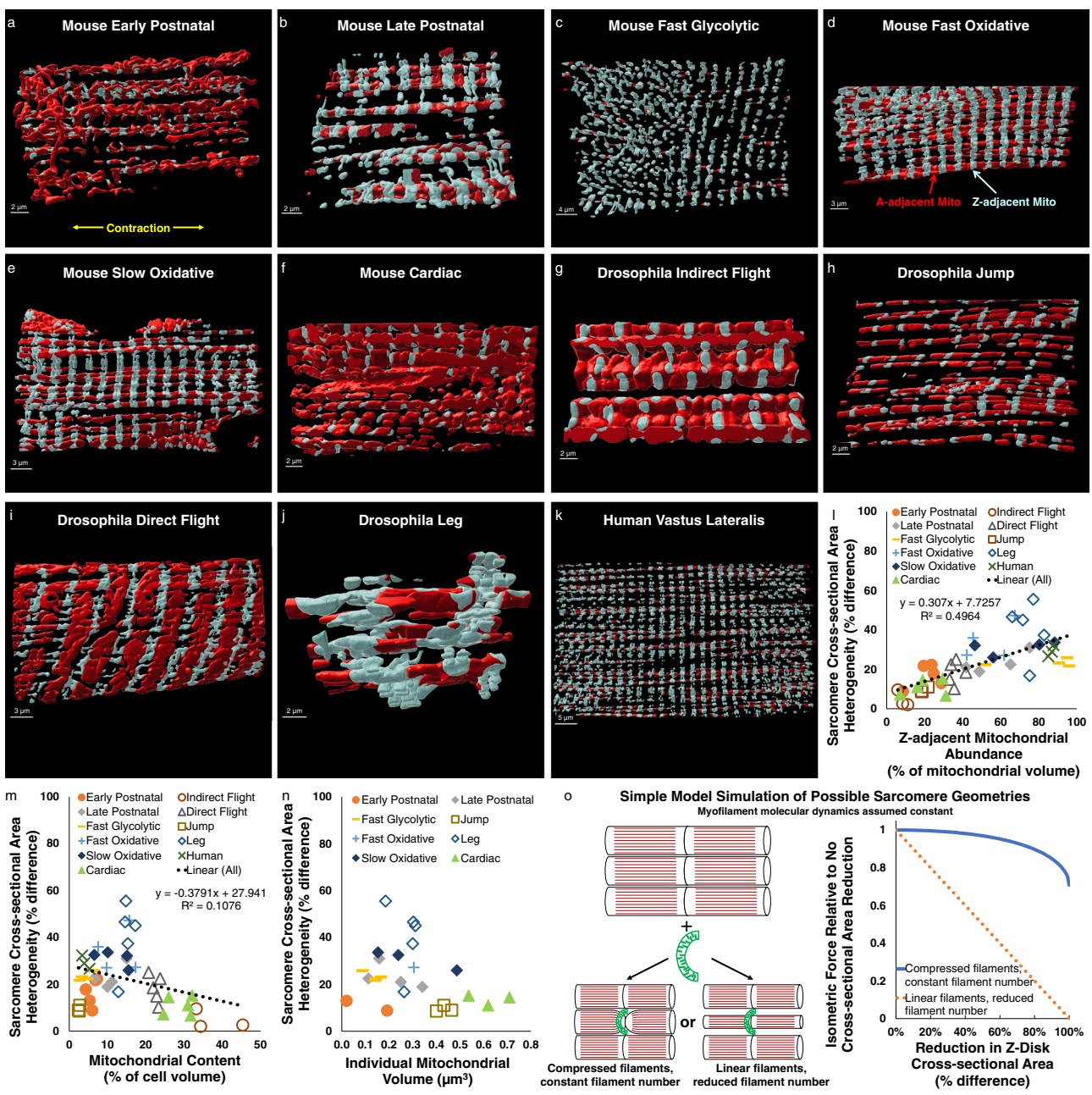

**Fig. 2 | Magnitude of intrasarcomere cross-sectional area heterogeneity is related to mitochondrial location. a–k** 3D rendering of representative mitochondrial network structures from eleven muscle types from mouse, *Drosophila*, and human tissues showing Z-adjacent regions (light blue) and A-adjacent regions (red). **l** Strong linear correlation (black dotted line, linear regression, *p* < 0.001) between Z-adjacent mitochondrial abundance and sarcomere cross-sectional area (CSA) heterogeneity. Individual points represent individual muscle cell values.

**m** Weak linear correlation (black dotted line, linear regression, *p* = 0.024) between mitochondrial content and sarcomere CSA heterogeneity. **n** Lack of correlation between individual mitochondrial volume and sarcomere CSA heterogeneity. **o** Simulations of simple sarcomere geometry models showing proportional reduction in isometric force when entire sarcomere CSA is reduced and minor reduction in force when only the Z-disk CSA is reduced.

myosin filament with the corresponding raw data across the length of a single sarcomere (Supplementary Fig. 2a–e, Supplementary Movie 11). Then we counted the number of missed segmentations (filament with missing skeleton) and over segmentations (two skeletons per filament or skeleton in interfilament space) at five points along the length of the myosin filaments representing both ends (*n* = 444, 441 filaments), the center (*n* = 448), and two intermediate points (*n* = 455, 448). Across all points, the total error rate was 5.14 ± 0.47% with the majority of the errors occurring due to missed segmentations at the sarcomere periphery (Supplementary Fig. 2f). Based on electron microscopy images such as Fig. 1a here as well as in the literature[1,15,42,56–58] (e.g., Fig. 1a in

Wang et al.[59]), we hypothesized that intrasarcomere CSA heterogeneity also leads to variability in filament structures within a single sarcomere where filaments near organelles at the sarcomere periphery are slightly curved while those in the center remain linear.

To test this hypothesis, we fit each segmented myosin filament to a straight line and then measured the deviation of each filament from linearity as a proxy for filament curvature (Supplementary Movie 12). Indeed, a large variation in myosin filament curvature can be observed within a single sarcomere (Fig. 3b, c). To quantify myosin filament curvature relative to intrasarcomere positioning, we performed machine learning segmentation of the major organelles which

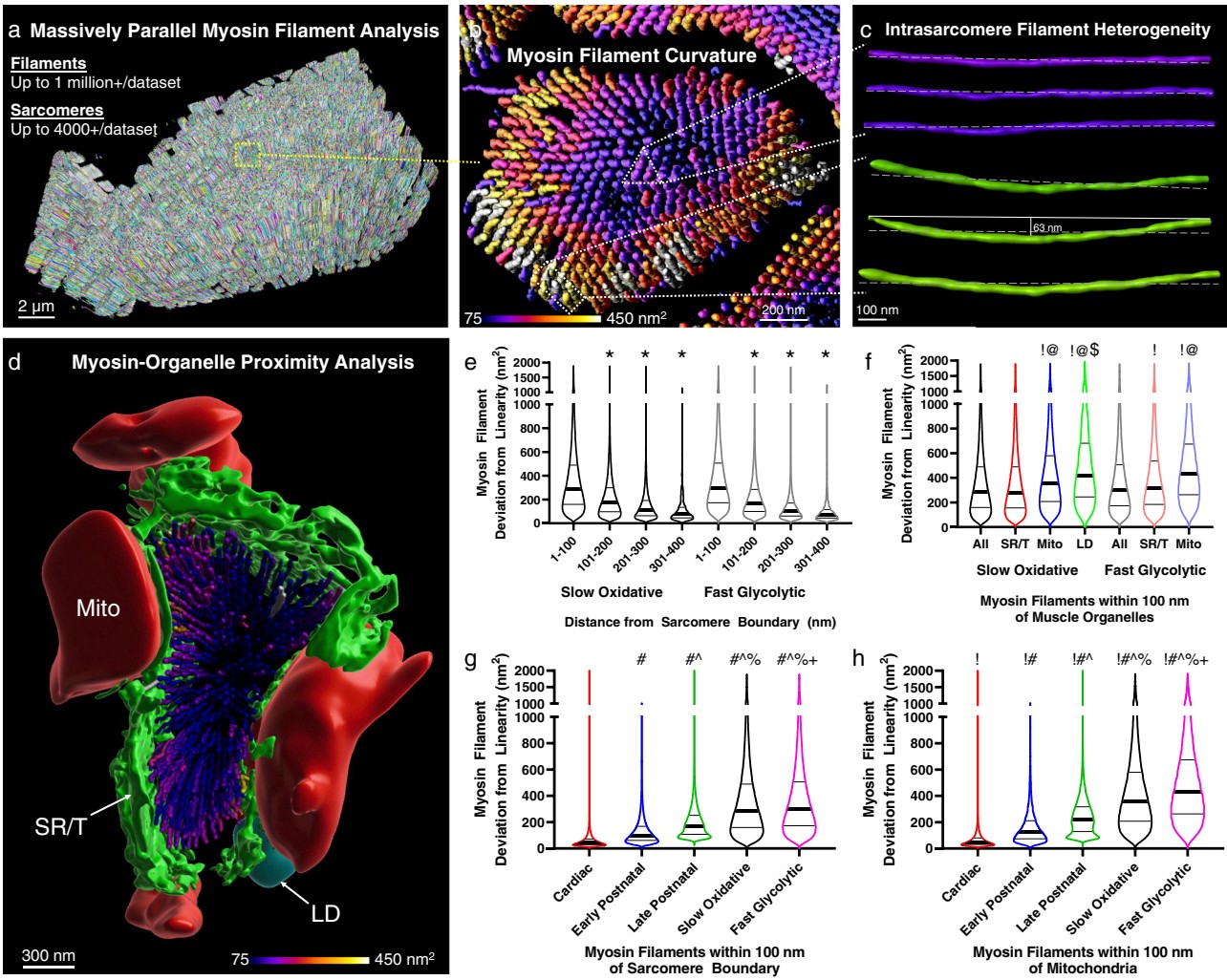

**Fig. 3 | Intrasarcomere myosin filament heterogeneity and organelle proximity. a** 3D rendering of 364,000+ myosin filaments (various colors) from a mouse late postnatal muscle. **b** 3D rendering of the myosin filaments from the highlighted sarcomere in (**a**). Filament color represents filament deviation from linearity. **c** 3D rendering of three myosin filaments from the sarcomere core in (**b**) (purple, upper) and three filaments from the periphery (green, lower) showing the variability in filament linearity. **d** 3D rendering of the myosin filaments from a single mouse cardiac sarcomere and the organelles which comprise the sarcomere boundary. Mitochondria (Mito, red), sarcotubular network (SR/T, green), and lipid droplets (LD, cyan) are shown. **e** Myosin filament deviation from linearity as a function of distance from the sarcomere boundary for Slow Oxidative and Fast Glycolytic fibers. **f** Myosin filament deviation from linearity for filaments within 100 nm of different muscle organelles. **g** Myosin filament deviation from linearity for filaments within 100 nm of the sarcomere boundary across cell types. **h** Myosin filament deviation from linearity for filaments within 100 nm of mitochondria across cell types. Thick black lines represent median values, thin black lines represent upper and lower quartile values. Width of the violin plot represents the relative number of filaments at a given value. *Significantly different from 1 to 100 nm. !Significantly different from All. @Significantly different from SR/T. $Significantly different from mitochondria. #Significantly different from Cardiac. ^Significantly different from Early Postnatal. %Significantly different from Late Postnatal. +Significantly different from Slow Oxidative. Two-sided, one-way ANOVA with a Tukey's HSD post hoc test. *N* values: Slow Oxidative-700,950 filaments; Fast Glycolytic-680,038; Cardiac-265,340; Early Postnatal-463,176; Late Postnatal-364,494.

surround and provide the boundaries for the sarcomeres (sarcoplasmic reticulum + t-tubules, SR/T; mitochondria, Mito; and lipid droplets, LD) (Fig. 3d, Supplementary Movie 13) and then measured the shortest distance between every myosin filament and each organelle.

In both slow oxidative and fast glycolytic muscles from the mature mouse, myosin filaments within 100 nm of the overall sarcomere boundary are significantly less linear than those more toward the center (Fig. 3e). Proximity to each individual organelle (SR/T, Mito, LD) also corresponds to greater myosin filament curvature than more distal filaments in slow oxidative and fast glycolytic muscles (Supplementary Fig. 3a–d). However, proximity to mitochondria and lipid droplets results in greater myosin filament curvature than proximity to SR/T or the overall sarcomere boundary (Fig. 3f). These data demonstrate that myosin filament structure is heterogeneous within mature mouse skeletal muscle sarcomeres with slightly curved filaments near

the periphery and more linear filaments within the core of the sarcomere. Additionally, proximity to larger organelles such as mitochondria and lipid droplets is associated with greater myosin filament curvature than smaller diameter organelles such as the SR/T.

Based on the cell type dependence of intrasarcomere CSA heterogeneity demonstrated above (Fig. 1) and the strong correlation with mitochondrial network orientation (Fig. 2), we hypothesized that myosin filament curvature would also be greatest in cell types with large proportions of perpendicularly oriented mitochondrial networks. Indeed, while myosin filaments located centrally within a sarcomere were more linear than peripheral myofilaments in mouse cardiac, early postnatal, and late postnatal muscles (Supplementary Fig. 3e) in addition to the mature mouse slow oxidative and fast oxidative muscles (Fig. 3e), the magnitude of myosin filament curvature was greater in the fast glycolytic, slow oxidative, and late postnatal muscles compared to the cardiac or early postnatal muscles (Fig. 3g).

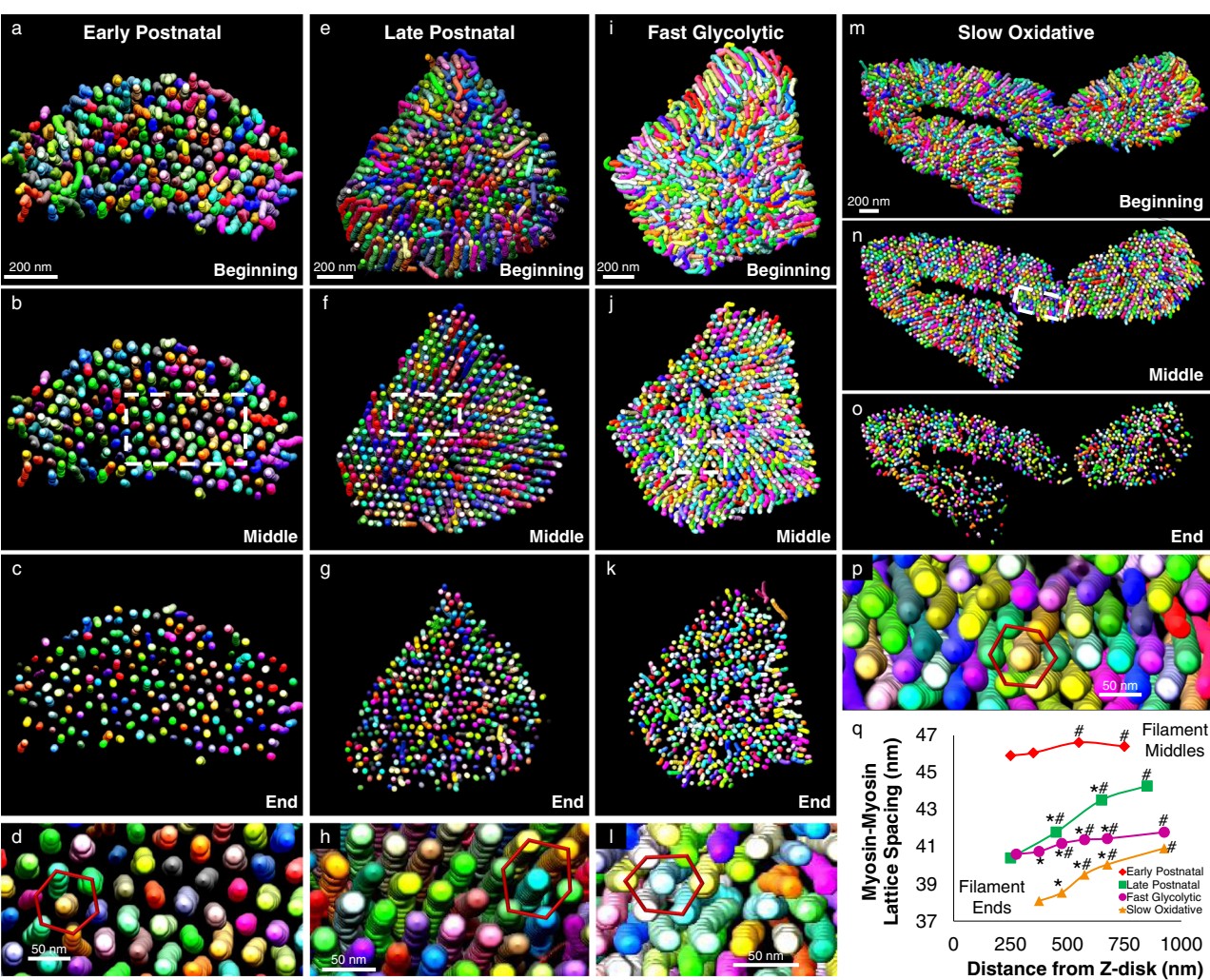

**Fig. 4 | Myosin lattice spacing intrasarcomere heterogeneity. a–p** 3D renderings of myosin filaments (various colors) within representative single sarcomeres from Early Postnatal (**a–d**), Late Postnatal (**e–h**), Fast Glycolytic (**i–l**), and Slow Oxidative (**m–p**) muscles showing the entire sarcomere (**a, e, i, m**), half the sarcomere (**b, f, j, n**), the ends of the sarcomere (**c, g, k, o**), and the hexagonal filament lattice from the middle of the sarcomere (**d, h, l, p**). **q** Myosin-myosin lattice spacing as a function of distance from the Z-disk for Early Postnatal (means shown as diamonds), Late Postnatal (squares), Fast Glycolytic (circles), and Slow Oxidative (triangles) muscles. *N* values: Early Postnatal-88, 88, 90, and 89 muscle cross-sections from center to end, respectively; Late Postnatal-342, 342, 340, and 333 muscle cross-sections; Fast Glycolytic-139, 139, 139, 139, 137, and 137 muscle cross-sections; Slow Oxidative-138, 138, 138, 138, and 93 muscle cross-sections. Standard error bars are smaller than mean symbols, thus not visible. *Significantly different from filament middles. #significantly different from filament ends. Two-sided, one-way ANOVA with a Tukey's HSD post hoc test.

Similarly, while proximity to mitochondria and lipid droplets was associated with greater myosin filament curvature than proximity to the SR/T or overall sarcomere boundary for all cell types (Fig. 3h, Supplementary Fig. 3e–j), the magnitude of myosin curvature for filaments nearby the larger organelles was also largely dependent on cell type (Fig. 3h, Supplementary Fig. 3h, i).

Together, these data show that myosin curvature in mouse muscles is highest in cell types with perpendicularly oriented mitochondrial networks, while within a given cell, myosin curvature is greater for filaments near mitochondria and lipid droplets compared to the SR/T. Thus, the presence of mitochondria adjacent to a sarcomere, particularly when oriented perpendicular to the contractile axis, influences the shape of the internal myofilaments within that sarcomere.

## Myosin lattice spacing varies along the sarcomere length

Force production within a sarcomere is based on physical interactions between actin and myosin, and, thus, is determined in part by the distances between myofilaments[60,61]. Within the A-band of the sarcomere, both actin and myosin are arranged in hexagonal lattice arrays where each myosin filament is surrounded by six other myosin filaments and also by six actin filaments which are closer to each myosin filament and form a smaller lattice[62,63]. Spacing within the myofilament lattices regulates sarcomere shortening velocity[64], length-tension relationships[52], cross-bridge kinetics[65,66], and advective-diffusive metabolite transport[67] with as small as a 1 nm lattice spacing change correlating with force production[68].

Thus, to better understand the potential functional implications of heterogeneous myofilament structures within a given sarcomere, we determined whether the lattice spacing between myosin filaments was variable among different regions of the mouse skeletal muscle sarcomere. Initial observations of single sarcomeres revealed the classic hexagonal myosin lattice within our FIB-SEM datasets although the spacing appeared to vary along the sarcomere length (Fig. 4a–p, Supplementary Movie 14). To quantify myosin center-to-center lattice spacing, we performed a 2D fast fourier transform (FFT) analysis of the myosin filament cross-sections at different regions along the length of the sarcomere. Whereas 2D FFT images of single sarcomeres show six bright spots surrounding the image center representing the hexagonal lattice (Supplementary Movie 15), performing a 2D FFT on the entire cross-section of a muscle dataset results in a circular profile around the

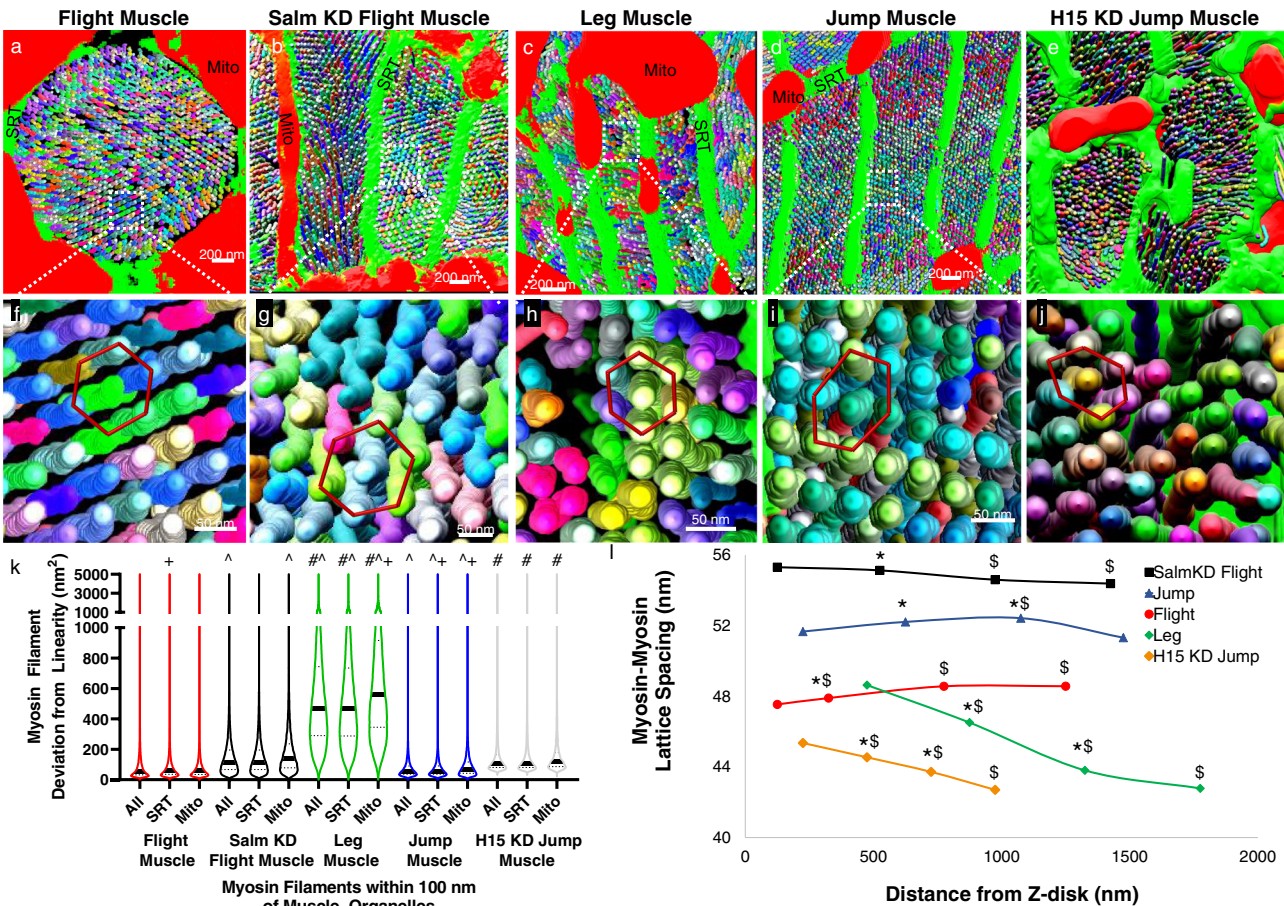

**Fig. 5 | Muscle specification factors *salm* and *H15* regulate myosin filament curvature and myosin-myosin lattice spacing in *Drosophila* muscle. a–e** 3D renderings of myosin filaments (various colors) and their proximal mitochondria (mito, red) and sarcoplasmic reticulum/t-tubules (SRT, green) in Drosophila flight (**a**), *salm* KD flight (**b**), leg (**c**), jump (**d**), and *H15* KD jump (**e**) muscles. **f–j** 3D rendering of myosin filament lattice in Drosophila flight (**f**), *salm* KD flight (**g**), leg (**h**), jump (**i**), and *H15* KD jump (**j**) muscles. **k** Myosin filament curvature for filaments within 100 nm of sarcomere boundary, SRT, and mitochondria. Thick black lines represent median values, thin dotted lines represent upper and lower quartile values. Width of the violin plot represents the relative number of filaments at a given value. #Significantly different from Jump muscles. ˄Significantly different

from Flight muscles. +Significantly different from All. N values: Leg-104,090 total filaments; Jump-992,756; Flight-1,282,854; *salm* KD Flight-338,765. H15 KD Jump-113,615. **l** Myosin-myosin lattice spacing as a function of distance from the Z-disk for Leg (means shown as diamonds), *salm* KD Flight (squares), Flight (circles), Jump (triangles), and *H15* KD Jump (diamonds) muscles. N values: Leg-38, 40, 36, and 38 muscle cross-sections from center to end, respectively; *salm* KD Flight-36, 36, 36, and 36 muscle cross-sections; Flight-60, 60, 60, and 58 muscle cross-sections; Jump-80, 79, 80, and 78 muscle cross-sections; *H15* KD Jump-63, 62, 62, and 60 muscle cross-sections. Standard error bars are smaller than mean symbols, thus not visible. *Significantly different from filament middles. $Significantly different from filament ends. Two-sided, one-way ANOVA with a Tukey's HSD post hoc test.

center due to the different orientations of the many sarcomeres within the field of view (Supplementary Fig. 4).

To assess lattice spacing in different regions of the sarcomere, we performed whole dataset FFT analyses after segmenting each myosin filament into separate 50 nm regions representing the filament centers, filament ends, and 2–4 intermediate points in between based on their distances from the Z-disk (Supplementary Fig. 4c, f). In early postnatal muscle, myosin lattice spacing varied by less than 1 nm along the sarcomere length with a $45.91 \pm 0.11$ nm lattice near the filament ends ($n = 89$ muscle cross-sections) and slightly larger $46.41 \pm 0.09$ nm spacing near the filament centers ($n = 88$) (Fig. 4q). Conversely, myosin lattice spacing in late postnatal, fast glycolytic, and slow oxidative muscles all varied by greater than 1 nm along the sarcomere length with larger spacing among the filament centers than for the filament ends (Fig. 4q). The late postnatal muscle myosin lattice was the most variable of the muscles assessed here and changed by almost 4 nm from $44.28 \pm 0.03$ nm ($n = 342$) in the sarcomere center to $40.40 \pm 0.09$ nm ($n = 333$) near the filaments ends. Myosin lattice spacing in the slow oxidative muscles varied by nearly 3 nm from $40.92 \pm 0.09$ nm ($n = 138$) in the center of the sarcomere to $38.09 \pm 0.10$ nm ($n = 93$) near the filament ends. Finally, in the fast

glycolytic muscles, myosin lattice spacing varied from $41.79 \pm 0.07$ nm ($n = 139$) near the sarcomere center down to $40.60 \pm 0.09$ nm ($n = 137$) near the filament ends. These data demonstrate intrasarcomere heterogeneity of myosin-myosin lattice spacing and suggest that the actin-myosin molecular interactions which govern muscle contraction may also vary along the length of a single sarcomere.

### Salm and H15 regulate myofilament interactions in drosophila

To determine whether intrasarcomere heterogeneity in myosin filament shape and lattice spacing is conserved in *Drosophila*, we performed a massively parallel myosin filament segmentation in the leg, jump, and flight muscles (Fig. 5). Both the jump and leg muscles have a tubular contractile apparatus comprised of connected myofibrillar networks[4] while flight muscles are fibrillar (Fig. 1i, j) with many individual myofibrils[39,69]. Conversely, both the jump and flight muscles have parallel mitochondrial networks (Fig. 2g, h) whereas the leg has grid-like mitochondrial networks[36,39] (Fig. 2j). Thus, we can directly compare muscles with similar contractile types but different mitochondrial network configuration (jump vs. leg) and muscles with similar mitochondrial network configurations but different contractile types (jump vs. flight).

Similar to the mouse muscles, myosin filaments in the interior of the sarcomere were more linear than peripheral filaments (Supplementary Fig. 5a) for each *Drosophila* muscle type confirming the evolutionarily conserved nature of intrasarcomere myosin filament shape heterogeneity. However, myosin filament curvature for peripheral filaments (1–100 nm from boundary) was higher in the leg muscles compared to the jump and flight muscles (Fig. 5k) consistent with the intrasarcomere CSA heterogeneity data in Fig. 1. These data suggest that mitochondrial network orientation rather than contractile type drives intrasarcomere variability in myosin filament shape in *Drosophila* muscles.

To further investigate whether mitochondrial network orientation regulates intrasarcomere variability, we knocked down (KD) flight muscle specification factor *salm*[29] in a muscle-specific manner (*Mef2-Gal4*) which converts the flight muscle to tubular with grid-like mitochondrial networks[36,39]. We recently showed that, in our hands, *salm* KD results in an ~60% loss of Salm mRNA in the thorax and a near complete loss of Salm protein in flight muscles as assessed by qPCR and immunofluorescent analyses, respectively[36]. Myosin filaments in s*alm* KD flight muscles were significantly less linear than wild type flight muscles, and filaments in proximity to mitochondria displayed even greater curvature than the overall sarcomere boundary (Fig. 5k). Thus, *salm*-mediated conversion of mitochondrial network configuration in *Drosophila* flight muscles leads to an increase in intrasarcomere myosin shape heterogeneity.

To investigate the impact of altering mitochondrial network configuration independent of contractile type, we assessed myosin filament shape heterogeneity in jump muscles lacking transcription factor *H15* (driven with *Mef2-Gal4*) which we recently showed causes a parallel to grid-like conversion of mitochondrial networks while maintaining the tubular nature of the contractile apparatus[36]. Myosin filaments within 100 nm of the sarcomere boundary, mitochondria, or SRT were all significantly less linear in *H15* KD jump muscles compared to their wild type counterparts (Fig. 5k). Overall, these data demonstrate that genetic manipulation of the muscle cellular design process to increase the proportion of mitochondria running perpendicular to the contractile axis near the Z-disk leads to greater myosin filament heterogeneity.

To determine how intrasarcomere variability in myosin filament shape affects myosin lattice spacing in *Drosophila* muscles, we performed 2D FFT analyses of myosin filament spacing in different regions along the length of the sarcomere (Supplementary Fig. 6) as described for mouse muscles above. In the flight muscles, myosin lattice spacing varied by ~1 nm across a 1125 nm distance along the sarcomere length with a $47.54 \pm 0.03$ nm lattice near the filament ends ($n = 58$ muscle cross-sections) and slightly larger $48.57 \pm 0.03$ nm spacing near the filament centers ($n = 60$) (Fig. 5l). In jump muscles, myosin lattice spacing was no different between the filament centers ($51.32 \pm 0.17$, $n = 80$) and ends ($51.67 \pm 0.14$, $n = 78$) but was greatest at 1075 nm from the Z-disk ($52.43 \pm 0.12$, $n = 79$) (Fig. 5l). These data suggest that the variability in myosin lattice spacing along the length of the sarcomere proceeds differently in tubular muscles than in fibrillar muscles or mouse skeletal muscles (Fig. 4q) where myosin filaments are furthest apart at the filament centers and closest together near the filament ends. Indeed, the tubular leg, *salm* KD flight, and *H15* KD jump muscles all have greater lattice spacing near the filament ends (leg: $48.64 \pm 0.23$, $n = 38$; *salm* KD flight: $55.31 \pm 0.13$, $n = 36$; *H15* KD jump: $45.36 \pm 0.18$, $n = 60$) than near the centers (leg: $42.79 \pm 0.28$, $n = 38$; *salm* KD flight: $54.38 \pm 0.10$, $n = 36$; *H15* KD jump: $42.70 \pm 0.21$, $n = 63$), though the magnitude of variability is greatest in the leg muscles (Fig. 5l).

This type of intrasarcomere variation in tubular muscle lattice spacing where myosin filaments are farthest apart at the filament ends is consistent with previous 3D renderings of scorpion tubular muscles (see Fig. 12 in ref. 70) though lattice spacing was not quantified nor related to mitochondrial location in that study. Additionally, it is important to note that loss of *salm* mediated an increase in myosin lattice spacing and a change in the direction of intrasarcomere lattice spacing variability compared to wild type flight muscles (Fig. 5l) whereas loss of *H15* led to decrease in lattice spacing in jump muscles. Overall, these data demonstrate that intrasarcomere heterogeneity is myosin lattice spacing is conserved in *Drosophila* muscles, though the direction of variability can differ from mouse muscles.

### Acute mitochondrial swelling induces myofilament curvature

The consistent relationship between mitochondrial location and intrasarcomere CSA and myosin filament shape heterogeneities described above suggests a localized coordination between the volumes occupied by the contractile and metabolic machineries during muscle cell development. We hypothesized further that acutely increasing mitochondrial volume would also require compensation by alterations in nearby sarcomere structures. Mitochondria are known to be highly sensitive to fixation conditions, and improper fixation can induce swelling in mitochondria without alteration to other organelles or cellular structures[71]. Thus, we assessed myosin filament linearity in mouse vastus lateralis muscle cells with localized mitochondrial swelling in response to suboptimal fixation.

In the fast-twitch glycolytic muscle fibers assessed, localized mitochondrial swelling can be clearly observed by a lack of cristae or other electron dense structures (green arrows in Fig. 6a) while the majority of mitochondrial volume maintains its thin, elongated, perpendicularly oriented network structure (Fig. 6c) with densely packed cristae (magenta arrows in Fig. 6a). Additionally, no apparent changes to sarcoplasmic reticulum or cell membrane structures were observed, and the hexagonal myosin lattice is maintained in these cells (Fig. 6a inset). The striking structural differences between swollen and normal mitochondria allowed for machine learning segmentation of both mitochondrial regions (Fig. 6b–e, Supplementary Movie 16), with swollen mitochondria having a 2.45-fold greater median mitochondrial diameter than normal mitochondria (Fig. 6f). Similar to the data reported above (Fig. 3f), myosin filaments near mitochondria were less linear than filaments near SRT (Fig. 6g). However, myosin filaments in proximity to the larger, swollen mitochondria displayed even greater curvature than filaments near the normal mitochondria (Fig. 6g). These data demonstrate that mitochondrial fixation artifacts can be easily detected and suggest that acute increases in mitochondrial volume are also compensated for by altering the shape of nearby sarcomeres.

## Discussion

The functional benefits of increasing mitochondrial content within a cell are generally well understood to increase the capacity for energy conversion, calcium buffering, ROS/metabolite signaling, and/or other processes in which mitochondria support cellular function[16,72–80]. However, in highly packed cells with limited cytosolic space, such as striated muscle cells, increasing the amount of mitochondria also requires a proportional loss of volume of other cellular structures. As such, there is a functional cost to adding a mitochondrion to a cell that is particularly important to consider when designing therapeutics which may increase mitochondrial content[81–84].

Here, we begin to investigate how striated muscle cells deal with this cost by evaluating how the structure of the sarcomere is affected by proximity to mitochondria across eleven total cell types from three species with a more than 18-fold range in mitochondrial content. We find that the CSA of the sarcomere can be variable along its length with the Z-disk being smaller than the middle of the A-band (Fig. 1), and that the magnitude of this intrasarcomere heterogeneity across cell types strongly correlates with the proportion of mitochondria located adjacent to the Z-disk (Fig. 2). These data suggest that placement of a mitochondrion oriented perpendicular to the contractile apparatus

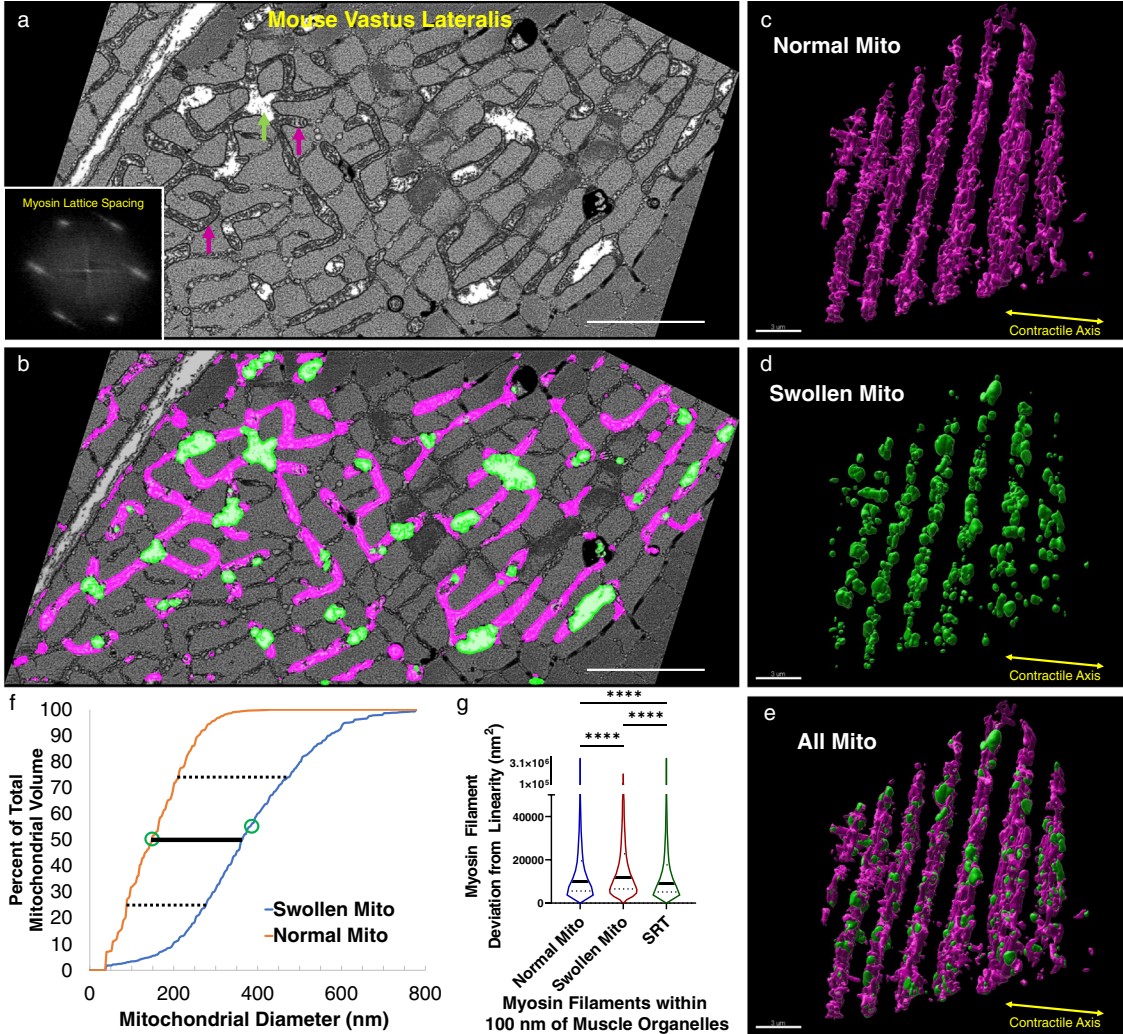

**Fig. 6 | Acute, localized mitochondrial swelling increases the curvature of nearby myosin filaments. a** Raw 2D FIB-SEM image of mouse vastus lateralis muscle with fixation induced swelling within several regions of the mitochondrial network. Green arrows: swollen mitochondria. Magenta arrows: normal mitochondria. Inset, average fast fourier transform image of raw FIB-SEM data across an entire single sarcomere showing hexagonal myosin lattice spacing. **b** Raw image from a overlaid with machine learning segmentation results for swollen (green) and normal (magenta) mitochondria. Scale bar: 1 μm (**c–e**) 3D renderings of normal (**c**), swollen (**d**), and all (**e**) mitochondria within a mouse vastus lateralis muscle. **f** Mitochondrial diameters for swollen (orange line) and normal (blue line)

mitochondria. Dotted lines: upper and lower quartile values, solid line: median value, open circles: mean value. Total volumes assessed: swollen−0.579 μm³, normal−1.864 μm³. **g** Myosin filament curvature for filaments within 100 nm of swollen mitochondria, normal mitochondria, and SRT. Thick black lines represent median values, thin dotted lines represent upper and lower quartile values. Width of the violin plot represents the relative number of filaments at a given value. $N = 46,400$, 11,289, and 74,418 myosin filaments, 2 cells for Normal, Swollen, and SRT, respectively. ****Significantly different ($p < 0.0001$) by one-way ANOVA (two-sided) with Tukey's post hoc test. Scale bars: 3 μm.

comes at the expense of sarcomere CSA, but that the functional cost may be minimized by limiting the reduction in CSA to near the Z-disk rather than the entire sarcomere (Fig. 2o).

By performing a massively parallel segmentation and analysis of over 4 million myosin filaments (Figs. 3, 5), we show that intrasarcomere CSA heterogeneity is achieved, at least in part, by the curvature of myosin filaments near the periphery of the sarcomere. Additionally, we find that myosin filaments in close proximity to large intrafibrillar organelles, such as mitochondria, have relatively greater curvature than filaments near smaller diameter organelles such as the sarcoplasmic reticulum. In turn, this variability among myosin filament structures within individual sarcomeres results in variable lattice spacing near the filament ends compared to the filament centers (Figs. 4, 5).

Alterations in myofilament lattice spacing throughout the entire sarcomere are known to directly affect several measures related to force production (shortening velocity, length-tension relationships, cross-bridge kinetics, etc.)[63–68]. However, it remains unclear how

variable lattice spacing within different regions of a single sarcomere influences force production. Moreover, while sarcomeric molecular dynamics simulations of linear actin and myosin filaments have suggested there is an optimal lattice spacing for force production[52], it is unknown how the different regions within the variable myosin lattices reported here compare to the theoretical optimal spacing.

As such, whether smaller lattice spacing closer to the filament ends results in increased force production relative to near the filament centers, perhaps due to increased probability of actin-myosin interactions[85,86] or potential stress induced activation of myosin in the compressed regions[87,88], or whether smaller lattice spacing leads to reduced force production, possibly by altering cross-bridge kinetics[65,66,89], is yet to be determined. With the current absence of methods to directly assess contractile forces with subsarcomeric precision, development of molecular dynamics models accounting for the variability in myofilament spatial relationships along the length of the sarcomere may provide key insights regarding the functional

implications of the intrasarcomere structural heterogeneities described here.

Variable lattice spacing within a sarcomere has been suggested to be a consequence of the transformation between the rhombic lattice at the Z-disk and the hexagonal lattice at the M-line[1]. This lattice mismatch would thereby result in oblique angles of the actin filaments and outward radial forces on the Z-disk during contraction which was proposed to be the cause of myofibril splitting during muscle growth[1,42]. However, comparing the intrasarcomere lattice spacing measurements from the four muscle types in Fig. 4 here with our previous assessment of myofibril splitting in the same muscle types[3] does not support this hypothesis as there is no direct positive or negative relationship between the two measurements (early postnatal muscle: 0.5 nm intrasarcomere lattice spacing variability, 27.6% of sarcomeres branch; late postnatal muscle: 3.8 nm, 12.8%; fast glycolytic muscle: 1.2 nm, 17.5%; slow oxidative muscle: 2.8 nm, 43.1%). Thus, while the heterogeneity in intrasarcomere myosin lattice spacing may well be the result of a mismatch between the M-line and Z-disk lattices, intrasarcomere variability in lattice spacing does not appear to be a cause of myofibril splitting.

In addition to myofilament lattice spacing, sarcomere length is also well known to play a critical role in contractile function[90–94]. Moreover, sarcomere length has been suggested to regulate sarcomere shape in contracted frog skeletal muscles with the diameter in the middle of the A-band being larger than at the Z-disk at sarcomere lengths where the I-band is visible[58]. Unfortunately, concomitant changes in the shape or location of the organelles located between the sarcomeres were not reported, and these data have been suggested to be artifacts of electron microscopy preparations[95].

However, in light of the data here, particularly the super-resolution images indicating smaller Z-disks compared to A-bands in live muscle cells at physiological sarcomere lengths (~2.1 μm) (Fig. 1l), together with previous electron microscopy and live cell studies in rat[96] and rabbit[97] cardiomyocytes showing that mitochondria become more compact during contraction and elongated during sarcomere lengthening, we would predict that sarcomere shape changes in accordance with sarcomere length are likely necessary to accommodate changes to the size and location of the organelles packed tightly between the sarcomeres. We are unaware of any simultaneous assessments of both sarcomere and organelle shapes at varying sarcomere lengths, and we did not specifically modulate sarcomere length in this work. Comparison of sarcomere lengths with sarcomere shape heterogeneity here did not reveal any consistent relationships across muscle types (Supplementary Fig. 1), though it is important to note that no muscles in the contracted or stretched state were assessed here. Due to the strong relationship between mitochondrial location and sarcomere shape established above (Fig. 2l), and the variability in mitochondrial location even among cells of the same type, we suggest that live cell studies capable of visualizing intrasarcomeric structures together with mitochondrial/organelle shape and location (e.g., Fig. 1l–n) while sarcomere length and/or contractile state are modulated dynamically will be needed to fully understand the integrated nature of the structure/function relationships between sarcomeres and their surrounding organelles.

In summary, by evaluating the 3D relationships among sarcomeres and their adjacent organelles across eleven muscle types from invertebrates to humans, we propose that where a mitochondrion is placed within the intramyofibrillar space influences the structure of the adjacent sarcomeres and can lead to heterogeneity of the cross-sectional area across individual sarcomeres in a cell-type specific manner. In turn, intrasarcomere CSA heterogeneity occurs together with curvature of the internal myosin filaments located at the periphery of the sarcomere with greater curvature of myofilaments located near mitochondria. This intrasarcomere heterogeneity in myosin filament shape thus results in variable lattice spacing among myosin

filaments in different regions of the sarcomere. While we are unable to clearly resolve actin filaments in our FIB-SEM datasets, it is likely that the intrasarcomere variability in myosin-myosin lattice spacing occurs concurrently with altered actin-myosin spacing, and thus, the force generating molecular interactions among actin and myosin are also likely variable at different regions within a sarcomere.

While it is possible that the chemical fixation procedures used here alter the lattice spacings observed in vivo[98–100], it does not seem likely that fixation would alter the lattice spacing variably in different regions of the sarcomere and in a cell type-dependent manner. Indeed, previous x-ray diffraction studies on live, intact frog sartorius muscle showed that both the A-band and Z-disk lattices are similarly sensitive to cellular osmolarity changes and that the Z-disk lattice area was smaller than the A-band lattice area at all osmolarities suggesting a barrel-like sarcomere shape[95,101] similar to the structures reported here. Additionally, the in vivo immersion fixation procedures used here maintain the circular profiles with physiological diameters of muscle capillaries[102] as well as the mitochondrial diameters observed in live cells[103] suggesting that the structures reported here are similar to the in vivo physiological state. Moreover, curved peripheral myofilaments, including actin, are also observed in cryo-preserved, non-chemically fixed sarcomeres[59] indicating that the structural heterogeneities within sarcomeres reported here are not simply artifacts of chemical fixation.

Further, we show here that the lattice spacing among mouse slow-twitch muscle myosin filaments are 38.09 to 40.92 nm in our in vivo fixed electron microscopy datasets, depending on which region of the sarcomere is assessed. This range compares favorably to live, intact mouse slow-twitch muscle cell bulk myosin lattice spacing values of 38.38 nm[104], 39.35 nm[105], and 41–43 nm[106] as assessed by X-ray diffraction ($d_{1,0}$ values multiplied by $2/\sqrt{3}$ to get interfilament spacings where necessary). Similarly, in mouse fast-twitch fibers, we find here that myosin lattice spacing varies between 40.60 and 41.79 nm which also compares favorably to intact, live cell x-ray diffraction measurements of 38.33 nm[107], 40.78 nm[108], and 40–42 nm[106]. These data indicate that the molecular interactions among myofilaments in our in vivo fixed muscles are within the physiological range of intact, live cell measurements. Thus, overall, these data indicate that the placement of a mitochondrion adjacent to the sarcomere not only alters the energetic support for muscle contraction but also influences the structure of the sarcomere down to the molecular interactions among myofilaments.

## Methods
### Mice
All procedures were approved by the National Heart, Lung, and Blood Institute Animal Care and Use Committee and performed in accordance with the guidelines described in the Animal Care and Welfare Act (7 USC 2142 § 13). 6–8 week old C57BL6/N mice were purchased from Taconic Biosciences (Rensselaer, NY) and fed ad libitum on a 12-h light, 12-h dark cycle at 20–26 °C and 22–70% humidity. Breeding pairs were setup, and progeny were randomly selected for each experimental group. Early Postnatal mice were from postnatal day 1 (P1) and Late Postnatal mice were from P14. Adult mice were 2–4 months of age. Animals were given free access to food and water and pups were weaned at P21. Due to difficulty using anogenital distance to reliably determine gender in P1 pups, we did not group mice depending on sex, but randomly used both male and female mice at all time points.

### Mouse muscle preparation
Mouse hindlimb and cardiac muscles were fixed and prepared for imaging as described previously[11]. Mice were placed on a water circulating heated bed and anesthetized via continuous inhalation of 2% isoflurane through a nose cone. Hair and skin were removed from the hindlimbs, and the legs were immersed in fixative containing 2% glutaraldehyde in 100 mM phosphate buffer, pH 7.2 in vivo for 30 min. For

cardiac fixation, the chest cavity was opened, and cardiac tissue was perfused through the apex of the left ventricle by slowly injecting 2 ml of relaxation buffer (80 mM potassium acetate, 10 mM potassium phosphate, 5 mM EGTA, pH 7.2) followed by 2 ml of fixative solution (2.5% glutaraldehyde, 1% formaldehyde, 120 mM sodium cacodylate, pH 7.2–7.4) through a syringe attached to a 30G needle. After initial fixation, the gastrocnemius, soleus, and/or left ventricles were then removed, cut into 1 mm$^3$ cubes, and placed into fixative solution for 1 h. After washing with 100 mM cacodylate buffer five times for 3 min at room temperature, the samples were placed in 4% aqueous osmium (3% potassium ferrocyanide, and 200 mM cacodylate) on ice for 1 h. The samples were then washed in bi-distilled H$_2$O five times for 3 min, and incubated for 20 min in fresh thiocarbohydrazide solution at room temperature. Afterwards, the samples were incubated for 30 min on ice in 2% osmium solution and then washed in bi-distilled H$_2$O five times for 3 min. Next, after incubating in 1% uranyl acetate solution overnight at 4 °C, the samples were washed in bi-distilled H$_2$O five times for 3 min and then incubated in lead aspartate solution (20 mM lead nitrate 30 mM aspartic acid, pH 5.5) at 60 °C for 20 min. After washing in bi-distilled H$_2$O at room temperature five times for 3 min, the samples were incubated sequentially in 20%, 50%, 70%, 90%, 95%, 100%, and 100% ethanol for 5 min each, and then incubated in 50% Epon solution and in 75% Epon solution for 3–4 h and overnight at room temperature, respectively. Epon solution was prepared as a mixture of four components: 11.1 ml Embed812 resin, 6.19 ml DDSA, 6.25 ml NMA, and 0.325 ml DMP-30. Next, the samples were incubated in fresh 100% Epon for one, one, and 4 h, sequentially. After removing excess resin using filter paper, the samples were placed on aluminum ZEISS SEM Mounts in 60 °C for 2–3 days. Then, using a Leica UCT Ultramicrotome (Leica Microsystems Inc., USA) that is equipped with Trimtool 45 diamond knife, the samples were trimmed, and then gold-coated using a sputter.

## Drosophila stocks and muscle preparation

W$^{1118}$ (BS# 3605) flies from the Bloomington Drosophila Stock Center were crossed on yeast corn medium (Bloomington Recipe) at 22 °C. *Mef2- Gal4* (III) (BS# 27390) was used to drive muscle-specific gene knockdown. RNAi lines for knockdown of *salm* (*UAS-salm RNAi*, V101052) and *H15* (*UAS-H15 RNAi*, V28415) were purchased from the Vienna Drosophila Resource Center. Muscles from 2–3 days old flies were dissected on standard fixative solution (2.5% glutaraldehyde, 1% formaldehyde, and 0.1 M sodium cacodylate buffer, pH 7.2) and processed for FIB-SEM imaging as described above.

## Focused ion beam scanning electron microscopy

Focus ion beam scanning electron microscopy (FIB-SEM) images were acquired using a ZEISS Crossbeam 540 with ZEISS Atlas 5 software (Carl Zeiss Microscopy GmbH, Jena, Germany) and collected using an in-column energy selective backscatter with filtering grid to reject unwanted secondary electrons and backscatter electrons up to a voltage of 1.5 keV at the working distance of 5.01 mm with a pixel size of 10 nm. FIB milling was performed at 30 keV, 2–2.5 nA beam current, and 10 nm thickness. Image stacks within a volume were aligned using Atlas 5 software (Fibics Incorporated) and exported as TIFF files for analysis. Some raw FIB-SEM datasets were previously used to assess mitochondria-organelle interactions[11,48], myofibril network connectivity[3,4], and/or endothelial cell/myocyte interactions[102].

## Image segmentation

Raw FIB-SEM image volumes were rotated in 3D so that the XY images within the volume were of the muscle cell cross-section. For segmentation of muscle mitochondria, lipid droplets, sarcotubular networks (SRT, sarcoplasmic reticulum + t-tubules), Z-disks, and A-bands, datasets were binned to 20 nm isotropic voxels, and semi-automated machine learning segmentation was performed using the Pixel

Classification module in Ilastik[50] as described previously[11]. Segmentation probability files were exported as 8-bit HDF5 files for import into ImageJ for subsequent analyses. Segmentation of individual mitochondria was performed by using Pixel Classification in Ilastik to generate outer mitochondrial membrane probabilities which were then loaded into the Multicut module in Ilastik for individual organelle separation as described previously[11].

For segmentation of individual myosin filaments, raw FIB-SEM data were upscaled to 5 nm isotropic voxels using bicubic interpolation within the Scale feature in ImageJ[109] resulting in 8-bit datasets of up to 160 Gb. A series of raw data images were loaded into complete volume memory in Thermo Scientific Avizo Software 2020.3 with the XFiber extension (Thermo Fisher Scientific, Waltham, MA) making sure to specify voxels were 5 nm. The XFiber extension permits segmentation of tightly packed cylindrical objects by first computing normalized cross correlation of the images against a hollow cylindrical template and by then tracing centerlines along filaments[110].

Cylinder correlation was performed using a template cylinder length of 100 nm, angular sampling of 5, mask cylinder radius of 20, outer cylinder radius of 19, and an inner cylinder radius of 0. Using a Windows 10 desktop PC with 64 logical processors (Intel Xeon Gold 6142 M), 2.0 TB of RAM, and an NVIDIA Quadro RTX8000 48 Gb GPU, cylinder correlation took 3–12 days depending on the size of the dataset and parameters chosen. Myosin filaments were then segmented from the resulting correlation and orientation fields using the Trace Correlation Lines module in Avizo. Minimum Seed Correlation (range 105–125) and Minimum Continuation (range 60–95) values within the Trace Correlation Lines module were varied per dataset using inspection of the correlation field images as guide for correlation values which best corresponded to myosin filaments in the raw data. Direction coefficient was set to 0.3, Minimum Distance was 20, Minimum Length was 500 nm, Search Cone Length, Angle, and Minimum Step Size were 100 nm, 30, and 10%, respectively. Trace Correlation Lines typically took 0.5–3 days depending on dataset size and parameters chosen. The resultant correlation lines were then converted with the Convert Geometry to Label module using the input raw dataset size parameters and saved as a 3D.raw file for import into ImageJ for subsequent analyses.

## Image analysis

Ilastik 8-bit HDF5 probability files for organelle and contractile structures were imported into ImageJ using the Ilastik plugin and made into binary files by using an intensity threshold of 128. Small segmentation errors were filtered out of the binary image volumes using the Remove Outliers plugin with block radius values from 3–10 and standard deviation values from 1.5 to 2.0. Z-disk CSA was assessed per half sarcomere sheet (Supplementary Movie 4) by first performing a Grouped Z maximum projection (Image-Stacks-Tools) at the predetermined sarcomere length. Because each Z-disk is part of two adjacent half sarcomeres, the maximum projection image was duplicated and the two images interleaved (Image-Stacks-Tools) together. Then the intensity of each image assessed using Plot Z-axis profile (Image Stacks) and the data copied and pasted into Excel (2016, Microsoft) for comparison to the corresponding A-band CSA values. A-band half sarcomere sheet CSA was assessed by first performing a Grouped Z average projection at half sarcomere length. Binary images were created from the average projection by normalizing local contrast (Plugins-Integral Image Filters) with a quarter sarcomere radius, 10 standard deviations, and marking the stretch and center boxes. The resultant image was masked by the A-band Grouped Z maximum projection at half sarcomere length and then intensity thresholded using the auto default value. Plot Z-axis profile was then used to assess the A-band intensity for each image, and the data copied and pasted to Excel where it was divided by the corresponding Z-disk intensity value yielding a relative CSA difference per half sarcomere sheet.

Z-disk adjacent mitochondrial abundance was determined by first separating the mitochondrial networks into two regions based on the proximity to the Z-disk. A Distance Transform 3D (Plugins-Process) was performed on the binary Z-disk image volumes and thresholded at a value 200 nm larger than the width of the I-band visible in the raw FIB-SEM datasets. This value was determined to encompass all perpendicularly oriented mitochondria based on analyses of the late postnatal and mature muscles in the mouse, leg muscles in *Drosophila*, and human muscles. A mask of the Z-adjacent regions was then created by subtracting (Process-Math) 254 from the thresholded Z-disk distance transform image, and the mask was multiplied by the binary mitochondrial segmentation image to create a Z-adjacent mitochondrial image volume. The total number of Z-adjacent mitochondrial pixels was determined from a Histogram (Analyze) of the entire stack and divided by the total number of mitochondrial pixels from the original binary mitochondrial image. Total mitochondrial content was determined by dividing the total number of mitochondrial pixels by the total number of cellular pixels. Mitochondrial diameter was assessed with the Local Thickness plugin[111] (Analyze-Local Thickness-Local Thickness (complete process)).

For myosin filament analyses, the.raw file from Avizo was imported into ImageJ (File-Import) making sure to enter the corresponding Width and Height pixel values and Number of images. Myosin filament deviation from linearity was assessed using the Particle Analyser[112] within the BoneJ[113,114] plugin for ImageJ (Plugins-BoneJ-Analyze) and selecting only the Moments of Inertia and Show Particle Stack options. The resultant data table and labeled filament image volume were saved for further analyses. The moment of inertia along the longest principal axis (I3) was then divided by the volume for each myosin filament to determine its deviation from linearity[115] by using a custom ImageJ macro to perform math operations within results tables. Visualization of filament deviation from linearity values was performed by using the Assign Measure to Label function within the MorphoLibJ[116] plugin (Plugins-MorphoLibJ-Label Images) using the labeled filament image and the I3/volume values from the results table.

For comparisons of filament deviation from linearity based on organelle proximity, image volumes of binary organelles segmented with 20 nm voxels were first upscaled (Image-Scale) to 5 nm voxels without interpolation. A Distance Transform 3D (Plugins-Process) was then performed on the mitochondria, lipid droplet, SRT, and total sarcomere boundary (mitochondria + lipid droplet + SRT) binary image volumes. The minimum distance between each myosin filament and the muscle organelles was then determined using the Intensity Measurements 2D/3D module within the MorphoLibJ plugin selecting the respective organelle distance transform image as Input, the myosin filament label image as Labels, and selecting only the Min Measurements box. The resultant tables were then saved and appended to the corresponding labels in the I3/volume table from above in Excel (for up to 1,048,576 labels) or SPSS (for more than 1,048,576 labels). Filaments that were >1 μm in length and did not overlap with any organelle were used for all proximity analyses.

Myosin filament center-to-center distances were assessed by 2D FFT analyses (Process-FFT) at different regions along the filament length. The myosin filament.raw file was imported into ImageJ as above and then multiplied (Process-Image Calculator) by the Z-disk distance transform image. The resultant filament distance from Z-disk image was then thresholded (Image-Adjust-Manual Threshold) to select a 50 nm region of the filament centers, near the filament ends, or at intermediate points in between and made into a binary image (Process-Binary-Make Binary). A maximum Z-projection (Image-Stacks-Tools-Grouped Z Project) was then performed for every 50 nm of the resultant binary image, and a 2D FFT was run for each projection image using a custom ImageJ macro to assess the periodicity of the myosin to myosin distances. The intensity of each resultant 2D FFT image was assessed using a custom ImageJ macro to iteratively select all pixels of

a given distance from the FFT center using a distance map from the FFT center (Plugins-Process-Exact Euclidean Distance Transform) and measuring their intensity (Plugins-MorphoLibJ-Analyze-Intensity Measurements 2D/3D). The resultant intensity profiles for each 2D FFT were then loaded into Excel, and the maximum intensity value for all distances between 30 and 60 nm was selected as the myosin to myosin filament distance for each respective image.

### Live muscle fiber isolation and imaging

The flexor digitorum brevis (FDB) muscle of mice genetically expressing mitochondrial outer membrane fluorophore TOMM20-mNeon[45] were dissected and immediately placed into Tyrode's buffer (pH 7.4, 10 mM HEPES, 137 mM sodium chloride, 4.5 mM potassium chloride, 0.5 mM magnesium sulfate, 0.5 mM potassium phosphate, 10 mM glucose, and 1.8 mM calcium chloride) and 3 mg/ml collagenase D (Roche) solution. The muscles were agitated in a water bath at 37 °C for 45–75 min. The Tyrode's plus collagenase solution was then removed, and the digested muscle was resuspended in Tyrode's. Individual muscle fibers were released by gentle trituration and allowed to settle to the bottom of the tube. The remaining Tyrode's solution was syphoned off, and cells were resuspended in 1 μM SPY555-Fast Act (Spirochrome, Switzerland) for 1 h prior to imaging.

FDB muscle cells were imaged with a Leica SP8 3X STED microscope with a Leica 100× (1.4 NA) STED White objective (Leica Microsystems, Inc., Wetzlar, Germany). TOMM20-mNeon and SPY555-Fast Act fluorescence was excited with a white-light laser set to 488 nm and 555 nm, respectively, while emission was collected by Leica HyD SMD time-gated PMTs from 500–545 nm and 565–640 nm, respectively. Huygens Professional software (version 19.1, Scientific Volume Imaging, Hilversum, the Netherlands) was used to deconvolve STED images based on idealized point spread functions, using the classic maximum likelihood estimation (CMLE) deconvolution algorithm.

### Geometric model of sarcomere isometric force production

Of the two simulated scenarios, the first scenario is a uniform CSA reduction with a proportional reduction of the number of myofilaments, keeping the lattice spacing unchanged. Assuming that the isometric force production of each filament also remains unchanged, then the total force is proportional to the total number of filaments. A reduction in the CSA directly results in an identical reduction in the isometric force. The second scenario simulates a myofibril of a circular cross-section, where the CSA tapers down gradually from the sarcomere center to the Z-disk in a smooth arc shape. It was assumed that the lattice spacing tapers down uniformly, and the tensile force of individual myofilaments was not altered by the compression. Each filament "pulled" on the Z-disk at a slightly off-perpendicular angle due to the variable curvature of the filament. This lead to a reduction of the axial force according to the sine factor of the filament insertion angle at the Z-disk. The overall effect on the force production was analytically derived by the integral of the axial force from each filament within the sarcomere bundle.

### Image rendering

Movies of 3D renderings of organelle and contractile structures were generated in Imaris 9.5 (Bitplane). Pictures of 3D renderings were created either using Imaris or the Volume Viewer plugin in ImageJ.

### Statistical analysis

Quantitative data was assessed using Excel 2016 (Microsoft), Prism 9.0.0 (Graphpad), or SPSS 28.0.0.0 (IBM) for all statistical analyses. All comparisons of means were performed using a two-sided, one-way ANOVA with a Tukey's HSD post hoc test. Linear regression analyses were performed in SPSS (Analyze-Regression-Linear) using default settings. A $p$ value < 0.05 was used to determine statistical significance.

**Reporting summary**

Further information on research design is available in the Nature Research Reporting Summary linked to this article.

## Data availability

The raw FIB-SEM datasets generated and/or analysed during the current study are available at (https://doi.org/10.5281/zenodo.5796264). Source data are provided with this paper.

## Code availability

ImageJ macros used for image analysis are provided in the Supplementary Information.

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

## Acknowledgements

We would like to acknowledge Eric Lindberg and Ye Sun of the NHLBI EM Core for assistance with collecting FIB-SEM datasets and Chris Combs and Daniela Malide of the NHLBI Light Microscopy Core for assistance with STED imaging. We thank Brian Caffrey (University of British Columbia), Sriram Subramaniam (University of British Columbia), and Luigi Ferrucci (National Institute of Aging) for providing access to the raw human muscle FIB-SEM datasets from the Baltimore Longitudinal Study of Aging. We are also grateful to Kenneth Campbell, Thomas Irving, James Sellers, Neil Billington, Kenneth Taylor, and Malcolm Irving for helpful discussions regarding the functional implications of the structures described here. This work was supported by the Division of Intramural Research of the National Heart Lung and Blood Institute and the Intramural Research Program of the National Institute of Arthritis and Musculoskeletal and Skin Diseases (1ZIAHL006221 to B.G.).

## Author contributions

P.K., H.A.P., B.G., and Y.K. prepped tissues for imaging. P.K., B.G., Y.K., and C.K.E.B. designed and B.G. and C.K.E.B. performed imaging experiments. A.S.H., P.T.A., T.B.W., and B.G. designed and performed

image analysis and B.G. created figures and videos. H.W. designed and performed computational modeling experiments. B.G. wrote and P.K., A.S.H., H.A.P., P.T.A., Y.K., T.B.W., H.W., C.K.E.B., and B.G. edited and approved the paper.

## Funding

## Competing interests
The authors declare no competing interests.
