## [Peer Review File · Nature Communications]

Mitochondrial Network Configuration Influences Sarcomere and Myosin Filament Structure in Striated MusclesEditorial Note: Parts of this Peer Review File have been redacted as indicated to remove third-party material where no permission to publish could be obtained.

REVIEWER COMMENTS

Reviewer #1 (Remarks to the Author):

Katti et al report a detailed analysis of myofibril and myosin filament morphology and how this changes in relation to the mitochondrial network. In particular the demonstration of changes observed with the knock down of salm in drosophila is particularly compelling. The reconstructions and videos are beautiful. This work is an important addition to the field and has the potential to change the way we think about muscle health and disease. I have a few major and few minor points. I also have a few comments and suggestions, which I hope will improve the accessibility of the paper for a diverse scientific audience.

Major points:

- 1) The authors state that for the heterogenous CSA model that the myofilaments curved slightly but that the myofilament number remained constant. Do the authors have data to support this assumption? They go on to reconstruct the myofilaments for example?
- 2) There are videos of the mitochondrial network and myofibril morphology in mouse and drosophila but no video for human. Is it possible to include a human video, of both for comparison?
- 3) Where automated segmentation is used have you checked the accuracy for segmentation, particularly for finer or complicated structures like the myofilaments and ER respectively, which you haven't reported previously?

Minor points:

- 1) The second half of the introduction is the conclusions of the paper which would be better in the discussion. I would suggest editing this to be the aims rather than the conclusions. I also feel that the background provided in the introduction is quite brief. What is known so far of mitochondria and myofibril arrangements across these three species. In particular are there important differences in mitochondrial network morphology that are likely to impact the findings and underpin your choices of muscles and species?
- 2) Throughout the results section some experiments are completed in one species and some in all 3. Can you make it clear for each mention of data in the text which species this comes from?
- 3) For people who aren't familiar with the different muscle types it would be beneficial to have either a paragraph in the introduction identifying the metabolic properties and features of the different muscles selected or to mention this throughout the results as different muscles are discussed.
- 4) The authors present data from human samples imaged for a previous study. However, because they are comparing these to the current samples it is important for the reader to know how these were processed and imaged, which is not at present included.
- 5) In figure 2 and elsewhere both cardiac and heart are used to refer to the mouse cardiac muscle (cardiac in f but heart in I, m and n) for consistency it would be better to use just one name throughout the paper and figures.
- 6) On page 11 line 277 the authors mention using previously published data and reference one example. Can they reference all data used please.
- 7) Supplementary figure 2 is performed in mouse, was this similar analysis completed in drosophila and human muscles?
- 8) The legend of Supplementary video 12 gives no mention to species?
- 9) In supplementary video 8 there appear to be central holes between the myofibril filaments in some of the myofibrils, what is located in these?

Reviewer #2 (Remarks to the Author):

Overall, this is an interesting and well-conducted study. Knowing the structural organization of the sarcomere and its intimate relationship to closely spaced mitochondria and other organelles is important and relevant. The finding that the sarcomere cross-sectional area varies along the length of the sarcomere, in a cell type-dependent manner, and that this might be related to its mitochondrial network configuration is interesting. The work appears overall well carried out and the results are clearly described. I have the following comments.

The sarcomere length at which the various muscle types should be shown and the effect of variation in sarcomere length on the structural parameters measured should be studied. Surely this must be important.

Similarly, the effect of chemical fixation on sarcomere structure should be addressed. Myofilament lattice spacings have been measured on intact muscle and how to the in vivo myofilament spacing compared to this measured in this study? What is the thick filament length in the fixed specimen? How would fixation-induced shrinkage impact the conclusions of this work? Which muscle type was used for the human study? This is important and should be revealed.

Reviewer #3 (Remarks to the Author):

In their manuscript, Dr Katti et al. took advantage of the capabilities of focused ion beam scanning electron microscopy, combined with advanced analytical approaches, to investigate possible interactions between the configuration of the mitochondrial network and the structure and organization of sarcomeres. To this end, the authors studied 11 muscles from 3 different species (drosophila, mouse, and human) with great variation in mitochondrial morphology, positioning and content as well as recruitment pattern. Using this elegant experimental design, they first provide data indicating that sarcomere cross-sectional area (CSA) varies along the sarcomere length in a cell type-dependent manner in flies and mice. They indeed convincingly show that some, but not all, muscles display fairly high heterogeneity in intrasarcomere CSA characterized by a smaller Z-disk CSA relative to the CSA at the sarcomere center. They also report that variation in sarcomere cross-sectional area is present in at least one human skeletal muscle (vastus lateralis), indicating that intrasarcomere CSA heterogeneity is a feature conserved from invertebrate to human skeletal muscles. Interestingly, the authors report that intrasarcomere CSA heterogeneity is not associated with total mitochondrial volume nor mitochondrial size but rather appears associated with the location of mitochondria between sarcomeres and is notably present when a high proportion of the mitochondrial network is localized right next to the Z-disk. The authors also provide data indicating that intrasarcomere CSA heterogeneity is achieved, at least in part, by curvature of the myosin filaments near the periphery of each sarcomere. Finally, the authors show that knocking down salm, a transcription factor recently shown to regulate mitochondrial network organization, in the flight muscle of *Drosophila melanogaster* leads to an increase in intrasarcomere myosin shape heterogeneity, suggesting that the alteration in mitochondrial network triggered by salm knockdown altered the organization of myosin filaments.

Overall, this manuscript is very well written and provides novel and important data with broad implications for our understanding of muscle bioenergetics and biomechanics. The concept of intrasarcomere CSA heterogeneity and its potential link to the organization of the mitochondrial network provides a potential framework whereby muscle force and power generation can be optimized in specific muscles without excessively compromising energy supply from mitochondrial oxidative phosphorylation. I believe that this manuscript will be quite influential as it furthers fundamental knowledge in muscle biology. The FIB-SEM images, 3D reconstructions and videos presented in this manuscript are also stunning. However, the following comments require consideration:

Major comments:

1- One of the main limitations of the present study is that it provides limited insight into the functional impact of intrasarcomere CSA heterogeneity on sarcomere and myofibril biomechanics. Assessing intrasarcomere CSA heterogeneity and myosin filament shape at various sarcomere lengths (shorten vs relaxed vs stretched) in at least one muscle displaying high intrasarcomere CSA heterogeneity would in my opinion provide valuable insights and further strengthen this manuscript.

2- The model provided in Fig. 20 for the assessment of the impact of a reduction in Z-disk CSA without a change in M-disk CSA is puzzling to me as it suggests that even a 40% reduction in Z-disk CSA has close to no impact on force generation. Such a negligible impact of a massive reduction in z-disk CSA makes me question the validity and usefulness of such model. Using this model, at what point does a reduction in Z-disk CSA majorly impacts force production? Also, I would assume that the impact of intrasarcomere CSA heterogeneity on force production likely vary depending on sarcomere length. Such potential relationship is never mentioned nor discussed in the manuscript.

3- The way the authors split their data in supplemental figure S1 is somewhat problematic. Indeed, by arbitrarily defining "low mitochondrial content cells" as cells with a mitochondrial volume <17.5% of the total cell volume and "high mitochondrial content cells" as cells with a mitochondrial volume >14.5% of the total cell volume, 9 cells end up being classified as both "high" and "low" mitochondrial content cells. It also looks like that these 9 cells are driving all significant correlations presented in Fig S1. I am not sure that any correlation between sarcomere CSA and mitochondrial content would remain if <14.5% was used as a cut off for cells with low mitochondrial content and if >17.5% was used as a cut off for cells with high mitochondrial content. The analysis and interpretation of data presented in Fig. 2m (which are identical to those presented in Fig S1) likely require more complex mathematical approaches than simple linear regressions.

4- The rationale for targeting only salm to alter the mitochondrial network organization appears unclear. Indeed, salm is known to control both mitochondrial and myofibril morphogenesis (Schönbauer et al., Nature, 2011 and Avellaneda et al., Nature Com., 2021, Katti et al., bioRxiv, 2021). By targeting salm it appears impossible to define whether changes in the mitochondrial network influences the ultrastructure of sarcomeres or vice-versa. In their recent preprint published in bioRxiv (Katti et al., bioRxiv, 2021), the authors report that knocking down H15 in jump and leg muscles converted the mitochondrial network to a more grid like arrangement without affecting the tubular nature of myofibrils. Assessing the impact of H15 knockdown in jump and leg muscles on intrasarcomere CSA and myosin heterogeneity would therefore appear as a better model to assess whether altering the mitochondrial network organization impacts the structure of sarcomeres and myosin filaments.

5- An additional way the authors might want to consider for altering the organization of mitochondrial network would be to overexpress Drp1. Drp1 overexpression was indeed shown to alter mitochondrial positioning in mouse skeletal muscle (Giovarelli et al., Cell Death and Differentiation, 2020). This might represent a way to further explore the link between intrasarcomere CSA heterogeneity and the organization of the mitochondrial network.

6- It is unclear as to why the authors only report the impact of salm knockdown on myosin heterogeneity and not on intrasarcomere CSA heterogeneity. Was intrasarcomere CSA heterogeneity altered in muscles with salm knockdown?

7- How was salm knockdown confirmed? What was the efficiency of the knockdown?

Minor comments:

1- The authors should indicate that human muscle samples were obtained from biopsies performed in the vastus lateralis. This information is not provided in the current version of the manuscript. Readers

are only referred to reference 19.

2- The authors should provide additional information on sample sizes when appropriate. For instance, the authors indicate that 3 human muscle cells were used to generate data presented in Fig. 1K. The authors should also indicate whether these 3 cells came from 1, 2 or 3 different participants. This comment applies to data collected in mice and flies.

3- The resolution of most graphs appears quite low.

4- Were male and female mice randomly used for P1 pups only or for all age groups?

5- The current title of the manuscript implies that sarcomere and myosin filament structures are dictated by the mitochondrial network configuration. While this is definitely a possibility, I do not think that the authors have the data to infer causality at this point. Their data do convincingly point towards a coordination between the mitochondrial network configuration and sarcomere and myosin filament structure. The authors might therefore want to update the title of their manuscript to more accurately reflect their findings.

6- Panel "(e)" in several figure legends was inadvertently converted to panel "€".

Response to Reviewers

REVIEWER COMMENTS

Reviewer #1 (Remarks to the Author):

Katti et al report a detailed analysis of myofibril and myosin filament morphology and how this changes in relation to the mitochondrial network. In particular the demonstration of changes observed with the knock down of salm in drosophila is particularly compelling. The reconstructions and videos are beautiful. This work is an important addition to the field and has the potential to change the way we think about muscle health and disease. I have a few major and few minor points. I also have a few comments and suggestions, which I hope will improve the accessibility of the paper for a diverse scientific audience.

We thank the reviewer for their constructive comments and suggestions which have helped to improve the clarity, reproducibility, and accessibility of our work.

Major points:

1) The authors state that for the heterogenous CSA model that the myofilaments curved slightly but that the myofilament number remained constant. Do the authors have data to support this assumption? They go on to reconstruct the myofilaments for example?

We thank the reviewer for these questions which are best answered from observing the 3D EM images and the myosin filament reconstructions in Figures 3-5. The curvature of the peripheral reconstructed filaments is shown in Figures 3b,c and 4e. Also, based on comment #3 from Reviewer #1 below, we have counted the total number of myosin filaments at different regions of a single sarcomere as part of the requested segmentation accuracy measurements. These data show that the total number of filaments varied by 1.6% or less between the middle of the sarcomere and the two ends.

2) There are videos of the mitochondrial network and myofibril morphology in mouse and drosophila but no video for human. Is it possible to include a human video, of both for comparison?

We thank the reviewer for this suggestion. We have now added a video of the human mitochondria/sarcomere interactions similar to the mouse and fly videos. The video shows the heterogeneity in sarcomere CSA with the smaller CSA regions corresponding to the location of perpendicularly oriented mitochondria much like in mature mouse muscle.

3) Where automated segmentation is used have you checked the accuracy for segmentation, particularly for finer or complicated structures like the myofilaments and ER respectively, which you haven't reported previously?

We thank the reviewer for this comment. We have now assessed the accuracy of the automated segmentation of finer structures by overlaying the myosin filament skeletons on the raw data for an entire sarcomere (~450 filaments) and counting the number of missed segmentations (no skeleton

present) and over segmentations (two or more skeletons on one filament or skeleton in interfilament space) at five different points along the length of the sarcomere. The total error rate (missed+over segmentations) was $5.14 \pm 0.47\%$ with the majority of the missed segmentation errors occurring at the periphery of the sarcomere. We have now added two supplemental videos showing the skeletons overlaid on the raw data for a single sarcomere as well as showing fully reconstructed filaments overlaid on the raw data across a much larger field of view in addition to adding a supplemental figure showing the skeleton overlays at the five regions assessed and the resultant data.

Minor points:

1) The second half of the introduction is the conclusions of the paper which would be better in the discussion. I would suggest editing this to be the aims rather than the conclusions. I also feel that the background provided in the introduction is quite brief. What is known so far of mitochondria and myofibril arrangements across these three species. In particular are there important differences in mitochondrial network morphology that are likely to impact the findings and underpin your choices of muscles and species?

We thank the reviewer for this suggestion. However, Nature Communications formatting guide (<https://www.nature.com/documents/ncomms-formatting-instructions.pdf>) requires the final paragraph of the introduction to summarize the results and conclusions of this work which is why we have formatted it this way.

Nonetheless, we have added two additional paragraphs to the introduction including the aim of this work as well as a brief introduction to the different muscle types studied here.

2) Throughout the results section some experiments are completed in one species and some in all 3. Can you make it clear for each mention of data in the text which species this comes from?

We thank the reviewer for this comment. We have now clarified which species are being discussed at the introduction of each data section.

3) For people who aren't familiar with the different muscle types it would be beneficial to have either a paragraph in the introduction identifying the metabolic properties and features of the different muscles selected or to mention this throughout the results as different muscles are discussed.

We thank the reviewer for this suggestion and have now added an additional paragraph to the introduction including discussion of the different properties of the many muscle types assessed here.

4) The authors present data from human samples imaged for a previous study. However, because they are comparing these to the current samples it is important for the reader to know how these were processed and imaged, which is not at present included.

We thank the reviewer for this concern. We have now added information to the results upon introduction of the human data that the datasets we received were imaged by FIB-SEM, have 30 nm pixels in all three directions, and are from vastus lateralis muscle biopsies.

5) In figure 2 and elsewhere both cardiac and heart are used to refer to the mouse cardiac muscle (cardiac in f but heart in l, m and n) for consistency it would be better to use just one name throughout the paper and figures.

We thank the reviewer for catching this. We have now changed all instances to cardiac.

6) On page 11 line 277 the authors mention using previously published data and reference one example. Can they reference all data used please.

We have now added five additional references here.

7) Supplementary figure 2 is performed in mouse, was this similar analysis completed in drosophila and human muscles?

The corresponding fly data was included as Supplemental Figure 4 in the original manuscript and is now Supplemental Figure 5. The human datasets we received do not have sufficient resolution to visualize or segment myosin filaments, thus, we are unable to provide any human myosin filament data here.

8) The legend of Supplementary video 12 gives no mention to species?

We have now clarified in the legend that this data is from mouse muscle.

9) In supplementary video 8 there appear to be central holes between the myofibril filaments in some of the myofibrils, what is located in these?

The holes are generally filled with sarcoplasmic reticulum membranes. This can now be seen in the new Supplemental Video 10 showing the myosin filament segmentations overlaid on the raw data.

Reviewer #2 (Remarks to the Author):

Overall, this is an interesting and well-conducted study. Knowing the structural organization of the sarcomere and its intimate relationship to closely spaced mitochondria and other organelles is important and relevant. The finding that the sarcomere cross-sectional area varies along the length of the sarcomere, in a cell type-dependent manner, and that this might be related to its mitochondrial network configuration is interesting. The work appears overall well carried out and the results are clearly described. I have the following comments.

We thank the reviewer for their careful review and constructive criticisms which have helped to improve this work. We hope that the three major additions we made to the paper (providing live cell support for intrasarcomere shape heterogeneity, evaluation of fly H15 KD jump muscles showing an increase in myosin filament curvature, and analysis of acute mitochondrial swelling resulting in a localized increase in myosin filament curvature) have significantly improved this work despite a lack

of specific investigation into the impact of variable sarcomere lengths for the reasons described below.

The sarcomere length at which the various muscle types should be shown and the effect of variation in sarcomere length on the structural parameters measured should be studied. Surely this must be important.

We agree that sarcomere length is likely to play an important role in altering the relationships between the shapes of the organelles in the intersarcomere space and the shapes of the sarcomeres. While we did not specifically control for or modulate sarcomere length in our studies on 11 different wild type muscles as well as now 2 different mutant fly muscles, the sarcomere lengths and accompanying sarcomere CSA heterogeneity values for all wild type muscle are now provided in Supplemental Figure 1a. No consistent relationship is apparent to us in these data, though this is not strong evidence of a lack of importance of sarcomere length. The lack of relationship may be explained by the fact that mitochondrial location appears to be a primary driver of intrasarcomere CSA heterogeneity, and the percentage of mitochondria located near the Z-disk is variable, even across cells of the same cell type. Thus, we believe the impact of sarcomere length is best assessed in a live cell system (e.g. see next response below) where the same sarcomeres and organelles can be imaged dynamically at different lengths along with concomitant measures of contractile and organelle function. Unfortunately, this is a considerable undertaking and beyond what we can complete within a reasonable timeframe for this manuscript. However, discussion of the potential importance of sarcomere length has now been added.

Similarly, the effect of chemical fixation on sarcomere structure should be addressed. Myofilament lattice spacings have been measured on intact muscle and how to the *in vivo* myofilament spacing compared to this measured in this study? What is the thick filament length in the fixed specimen? How would fixation-induced shrinkage impact the conclusions of this work?

We thank the reviewer for these questions. To address the concerns about fixation, we have now performed STED super resolution microscopy on live mouse muscle cells labeling mitochondria, Z-disks, and part of the A-band (Figure 1l-o). While the intersarcomere space between the Z-disks near where mitochondria are located can clearly be resolved in these images, we cannot clearly resolve the intersarcomere space between the A-bands. These results are consistent with our EM images where the intersarcomere space between Z-disks where SR doublets are located (black arrows in Figure 1d,h) can be up to 300 nm or well within the resolution limits of STED microscopy. Conversely, the intersarcomere space between A-bands in our EM images is ~50 nm or less where longitudinal SR is located (yellow arrow in Figure 1f) and undetectable where no SR is present (blue arrow in Figure 1f), thus likely beyond the resolution limits of our STED images. These new live cell data suggest that adjacent sarcomeres are closer together at the A-band than they are at the Z-disk which indicates that our conclusions regarding sarcomere structural heterogeneities driven by organelles in the intersarcomere space based primarily on EM data are not solely due to fixation artifacts.

Additionally, as pointed out by the reviewer, the potential for fixation artifacts during electron microscopy preparations is always a concern. Fixation is classically done either *in situ* through perfusion of a fixative through the vasculature or *ex vivo* by removal of tissue and immersion in fixative. To avoid the known issues with each of these methods, we perform our mouse skeletal

muscle fixations *in vivo* by removing the overlying skin and fascia and dipping the hindlimb in fixative for 15-30 minutes. The muscles are then carefully excised for further processing and the live animal euthanized. Using this protocol previously, we showed that mitochondrial diameters in skeletal muscle 3D electron microscopy datasets are no different than those imaged in live cells by super resolution microscopy (Glancy et al. Nature, 2015). More recently, we demonstrated that capillaries within the muscle tissue maintain their open, circular nature with physiological diameters using this *in vivo* fixation approach (Kim et al. Cardiovascular Research, 2020). At the myofilament level, we show here that the hexagonal lattice spacings among mouse slow-twitch muscle myosin filaments are 38.09 to 40.92 nm in our *in vivo* fixed electron microscopy datasets, depending on which region of the sarcomere is assessed. This range compares favorably to live, intact mouse slow-twitch muscle cell bulk myosin lattice spacing values of 38.38 (Ma et al. J Physiol, 2020), 39.35 (Kiss et al. PNAS, 2018), and 41-43 nm (Zappe and Maeda, J Mol Biol, 1985) as assessed by x-ray diffraction ($d_{1,0}$ values multiplied by $2/\sqrt{3}$ to get interfilament spacings where necessary). Similarly, in mouse fast-twitch fibers, we find here that myosin lattice spacing varies between 40.60 and 41.79 nm which also compares favorably to intact, live cell x-ray diffraction measurements of 38.33 (Hill et al. Elife, 2021), 40.78 (Song et al. PNAS, 2021), and 40-42nm (Zappe and Maeda, J Mol Biol, 1985). For reference, live cell x-ray diffraction studies have been more commonly performed in permeabilized cells rather than intact cells and generally increases lattice spacing by more than 5 nm (Ma et al. J Physiol, 2020; Song et al. PNAS, 2021). Lattice spacings in permeabilized cells can then be returned to the physiological range by using dextran, for example, to compress the filaments (Cazorla et al. Circ Res, 2001; Irving et al. Biophys J, 2011; Fukuda et al. Pflugers Archiv, 2005). However, the lattice spacings in our *in vivo* fixed muscles are already within the physiological range of intact, live cell measurements as described above.

Finally, the curvature of peripheral myofilaments (both actin and myosin), which provides the physical mechanism for heterogeneous lattice spacing within a sarcomere, can be observed in non-chemically fixed sarcomeres visualized by cryo-electron tomography (Figure 1a from Wang et al. Cell, 2021 included below), though this was not discussed in that work. Moreover, it is unclear to us how the cell type specificity of intrasarcomere structural heterogeneity and the strong association with mitochondrial location across eleven muscle types in three species could be explained by fixation artifacts. While it is possible that fixation may have an effect on the precise quantitative measurements here, for all of the reasons mentioned above, including the new live cell data, we do not believe fixation has a considerable impact on any of the conclusions of this work. This discussion has now been added to the manuscript.

[redacted]

Which muscle type was used for the human study? This is important and should be revealed.

We thank the reviewer for this comment. Biopsies from vastus lateralis muscle were the source of the human data. This has now been added to the text upon first introduction of the human data.

Reviewer #3 (Remarks to the Author):

In their manuscript, Dr Katti et al. took advantage of the capabilities of focused ion beam scanning electron microscopy, combined with advanced analytical approaches, to investigate possible interactions between the configuration of the mitochondrial network and the structure and organization of sarcomeres. To this end, the authors studied 11 muscles from 3 different species (Drosophila, mouse, and human) with great variation in mitochondrial morphology, positioning and content as well as recruitment pattern. Using this elegant experimental design, they first provide data indicating that sarcomere cross-sectional area (CSA) varies along the sarcomere length in a cell type-dependent manner in flies and mice. They indeed convincingly show that some, but not all, muscles display fairly high heterogeneity in intrasarcomere CSA characterized by a smaller Z-disk CSA relative to the CSA at the sarcomere center. They also report that variation in sarcomere cross-sectional area is present in at least one human skeletal muscle (vastus lateralis), indicating that intrasarcomere CSA heterogeneity is a feature conserved from invertebrate to human skeletal muscles. Interestingly, the authors report that intrasarcomere CSA heterogeneity is not associated with total mitochondrial volume nor mitochondrial size but rather appears associated with the location of mitochondria between sarcomeres and is notably present when a high proportion of the mitochondrial network is localized right next to the Z-disk. The authors also provide data indicating that intrasarcomere CSA heterogeneity is achieved, at least in part, by curvature of the myosin filaments near the periphery of each sarcomere. Finally, the authors show that knocking down *salmon*, a transcription factor recently shown to regulate mitochondrial network organization, in the flight muscle of *Drosophila melanogaster* leads to an increase in intrasarcomere myosin shape heterogeneity, suggesting that the alteration in mitochondrial network triggered by *salmon* knockdown altered the organization of myosin filaments.

Overall, this manuscript is very well written and provides novel and important data with broad

implications for our understanding of muscle bioenergetics and biomechanics. The concept of intrasarcomere CSA heterogeneity and its potential link to the organization of the mitochondrial network provides a potential framework whereby muscle force and power generation can be optimized in specific muscles without excessively compromising energy supply from mitochondrial oxidative phosphorylation. I believe that this manuscript will be quite influential as it furthers fundamental knowledge in muscle biology. The FIB-SEM images, 3D reconstructions and videos presented in this manuscript are also stunning. However, the following comments require consideration:

We thank the reviewer for their thorough assessment of our manuscript and the constructive suggestions on how to improve our work. We hope the reviewer will find that the three significant additions we made to the paper (evaluation of H15 KD jump muscles showing an increase in myosin filament curvature, analysis of acute mitochondrial swelling resulting in a localized increase in myosin filament curvature, and providing live cell support for intrasarcomere shape heterogeneity) have resulted in a significant enhancement of this work despite a lack of data directly assessing the impact of varying sarcomere length.

Major comments:

1- One of the main limitations of the present study is that it provides limited insight into the functional impact of intrasarcomere CSA heterogeneity on sarcomere and myofibril biomechanics. Assessing intrasarcomere CSA heterogeneity and myosin filament shape at various sarcomere lengths (shorten vs relaxed vs stretched) in at least one muscle displaying high intrasarcomere CSA heterogeneity would in my opinion provide valuable insights and further strengthen this manuscript.

We agree with the reviewer that linking sarcomere shape heterogeneity to a functional impact is critical. While there appear to be no current methods to assess mechanical forces with subsarcomeric or even sarcomeric resolution within whole muscle cells allowing for direct evaluation of the functional impact of intrasarcomere CSA heterogeneity, the critical role of myofilament lattice spacing on force production has been relatively well characterized across a variety of muscle systems. Thus, myosin lattice spacing analyses across nine muscle types were performed in order to demonstrate a likely functional impact of the intrasarcomere shape heterogeneities defined here. We also agree that varying sarcomere length is likely to have an effect on not only sarcomere shape, but also intersarcomere space organelle shapes. As also stated to Reviewer #2 above, we believe the impact of sarcomere length is best studied in a dynamic, live cell system where the same sarcomere and intersarcomere space structures can be assessed across time due in part to the large variability in relative mitochondrial location, a major driver of CSA heterogeneity, across cells of the same type. Additionally, while the impact of sarcomere length on contractile function is well known, how changing sarcomere length affects the shape and function of the intersarcomere space organelles in skeletal muscles is not well understood. Thus, an integrated analysis of the impact of varying sarcomere length (and contractile state) on both contractile and organelle structure and function is much needed. However, we feel this should be done as a dedicated study and is beyond what we can feasibly complete within the timeline of this response. We have, however, added all the sarcomere lengths for each muscle assessed as well as discussion on the potential impact of sarcomere length on the relationships defined here.

2- The model provided in Fig. 2O for the assessment of the impact of a reduction in Z-disk CSA without a change in M-disk CSA is puzzling to me as it suggests that even a 40% reduction in Z-disk CSA has close to no impact on force generation. Such a negligible impact of a massive reduction in z-disk CSA makes me question the validity and usefulness of such model. Using this model, at what point does a reduction in Z-disk CSA majorly impacts force production? Also, I would assume that the impact of intrasarcomere CSA heterogeneity on force production likely vary depending on sarcomere length. Such potential relationship is never mentioned nor discussed in the manuscript.

We thank the reviewer for these questions and have now updated the figure to include simulation of theoretical Z-disk CSA reduction up to 100%. We have also further clarified that these simulations are simply based on the number and angle of the myosin filaments and do not account for interactions with actin. Geometrically, as the outer filaments curve near the Z-disks, the perpendicular force decreases as the cosine function of the tilt angle of the filaments from the perpendicular direction. The cosine of the tilt angle is a second order function of the Z-disk diameter reduction; the diameter reduction is roughly half the area reduction. The combination of the two factors lead to a small effect at Z-disk reductions within the ranges measured in our muscles. By pure geometric consideration, using a typical value of (Z-disk diameter/inter Z-disk distance) = 0.6, at the extreme theoretical case where the CSA at the Z-disk shrinks down to zero, the filament tilt angle at the Z-disk would range from 0 degrees at the center to 57 degrees at the edge of the bundle, resulting in a 30% loss of total force. However, physically, the internal molecular dynamics of the filaments will likely be significantly disrupted at high levels of compression near the Z-disk, which would overtake the geometric effect and be the main factor that alters force generation. As stated in the response to the above comment, we now have added a paragraph discussing the potential impact of sarcomere length on the findings reported here.

3- The way the authors split their data in supplemental figure S1 is somewhat problematic. Indeed, by arbitrarily defining “low mitochondrial content cells” as cells with a mitochondrial volume <17.5% of the total cell volume and “high mitochondrial content cells” as cells with a mitochondrial volume >14.5% of the total cell volume, 9 cells end up being classified as both “high” and “low” mitochondrial content cells. It also looks like that these 9 cells are driving all significant correlations presented in Fig S1. I am not sure that any correlation between sarcomere CSA and mitochondrial content would remain if <14.5% was used as a cut off for cells with low mitochondrial content and if >17.5% was used as a cut off for cells with high mitochondrial content. The analysis and interpretation of data presented in Fig. 2m (which are identical to those presented in Fig S1) likely require more complex mathematical approaches than simple linear regressions.

We thank the reviewer for these comments. We originally chose to split the groups in this figure based on its data distribution with clear groupings below 14.5%, above 17.5%, and in between, with the 14.5-17.5% group being included on both sides due to a lack of clear separation point and also being the peak of the biphasic distribution. By removing this group, the significant linear relationships for the high mitochondrial content panels (b and d) still remain while the significant relationship in panel a is lost. However, since this analysis of high and low mitochondrial content cells is observational in nature, tangential to the focus of this paper, and does not impact any of the primary

findings, we have now removed this supplemental figure from the paper.

4- The rationale for targeting only salm to alter the mitochondrial network organization appears unclear. Indeed, salm is known to control both mitochondrial and myofibril morphogenesis (Schönbauer et al., Nature, 2011 and Avellaneda et al., Nature Com., 2021, Katti et al., bioRxiv, 2021). By targeting salm it appears impossible to define whether changes in the mitochondrial network influences the ultrastructure of sarcomeres or vice-versa. In their recent preprint published in bioRxiv (Katti et al., bioRxiv, 2021), the authors report that knocking down H15 in jump and leg muscles converted the mitochondrial network to a more grid like arrangement without affecting the tubular nature of myofibrils. Assessing the impact of H15 knockdown in jump and leg muscles on intrasarcomere CSA and myosin heterogeneity would therefore appear as a better model to assess whether altering the mitochondrial network organization impacts the structure of sarcomeres and myosin filaments.

We thank the reviewer for this suggestion and have now compared H15 KD jump muscles with the wild type jump muscles. These data show that H15 KD results in greater myosin filament curvature and more variability in lattice spacing indicating that altering mitochondrial network configuration without changing contractile type also alters sarcomere shape. These results have been added to Figure 5.

5- An additional way the authors might want to consider for altering the organization of mitochondrial network would be to overexpress Drp1. Drp1 overexpression was indeed shown to alter mitochondrial positioning in mouse skeletal muscle (Giovarelli et al., Cell Death and Differentiation, 2020). This might represent a way to further explore the link between intrasarcomere CSA heterogeneity and the organization of the mitochondrial network.

We appreciate the reviewer for providing multiple possible ways to more directly investigate the impact of mitochondrial organization on sarcomere structure. We have decided to take the reviewer's first suggestion to compare H15 KD jump muscles as mentioned above.

6- It is unclear as to why the authors only report the impact of salm knockdown on myosin heterogeneity and not on intrasarcomere CSA heterogeneity. Was intrasarcomere CSA heterogeneity altered in muscles with salm knockdown?

We thank the reviewer for this question. Unfortunately, salm KD in the flight muscles (and H15 KD in the jump muscles) causes non-SR mediated holes in the middle of the majority of Z-disks (see Supplemental Videos 5,6,9 in Ajayi et al. Nat Comms, 2022) without corresponding holes in the myosin lattice. These holes inherently make the Z-disk CSA smaller than the A-band CSA even in cases where the shape of the sarcomere is uniform along its length, thus invalidating the measure of relative A-band/Z-disk CSA as a measure of sarcomere shape heterogeneity in these muscles. Some holes in the Z-disk are also present in the eleven wild type muscles from flies, mice, and humans (see new Supplementary Video 5 here), but these holes are mediated by the SR which fills the holes and continues along the length of the sarcomere thus displacing both the Z-disk and A-band, maintaining the validity of the sarcomere shape heterogeneity measure in these sarcomeres.

7- How was salm knockdown confirmed? What was the efficiency of the knockdown?

We thank the reviewer for pointing out this missing info. In our hands, salm transcript levels are reduced by ~60% in the thoraces of the salm KD flies as assessed by qPCR, and this resulted in a near complete loss of salm protein in flight muscles assessed by immunofluorescence (Katti et al. bioRxiv, 2021). This info has now been added upon the introduction of the salm KD data in the Results.

Minor comments:

1- The authors should indicate that human muscle samples were obtained from biopsies performed in the vastus lateralis. This information is not provided in the current version of the manuscript. Readers are only referred to reference 19.

We thank the reviewer for this suggestion and have now added this information.

2- The authors should provide additional information on sample sizes when appropriate. For instance, the authors indicate that 3 human muscle cells were used to generate data presented in Fig. 1K. The authors should also indicate whether these 3 cells came from 1, 2 or 3 different participants. This comment applies to data collected in mice and flies.

We thank the reviewer for this suggestion. We have now indicated in the Figure 1 legend that the three human cells come from three different people as well as included similar info for the mouse and flies.

3- The resolution of most graphs appears quite low.

We apologize for this error which likely occurred during file conversion. We have now provided all main figures as individual files at full resolution in addition to within the manuscript file.

4- Were male and female mice randomly used for P1 pups only or for all age groups?

We thank the reviewer for this question. Both male and female mice were used randomly for all age groups. This has now been clarified in the methods.

5- The current title of the manuscript implies that sarcomere and myosin filament structures are dictated by the mitochondrial network configuration. While this is definitely a possibility, I do not think that the authors have the data to infer causality at this point. Their data do convincingly point towards a coordination between the mitochondrial network configuration and sarcomere and myosin filament structure. The authors might therefore want to update the title of their manuscript to more accurately reflect their findings.

We thank the reviewer for this suggestion. We have now added an additional experiment where we show that acute swelling of mitochondria induces increased curvature of nearby myosin filaments. Thus, these new data demonstrate that acute alterations in mitochondrial morphology can impact the shape of the contractile machinery, which is reflected by the current title.

6- Panel "(e)" in several figure legends was inadvertently converted to panel "€".

We thank the reviewer for catching this error which has now been corrected.

REVIEWERS' COMMENTS

Reviewer #1 (Remarks to the Author):

Thank you to the authors for addressing my comments and those of the other reviewers. I believe this has improved the manuscripts readability and accessibility and I am happy that it should be published in Nature Communications. I look forward to seeing this work published.

Reviewer #3 (Remarks to the Author):

The additional analyses and experiments performed by the authors satisfactorily addressed my main comments and concerns. I commend the authors for this body of work.